# *SF3B1* hotspot mutations confer sensitivity to PARP inhibition by eliciting a defective replication stress response

Philip Bland [1], Harry Saville[1], Patty T. Wai[1], Lucinda Curnow[2], Gareth Muirhead[1], Jadwiga Nieminuszczy [2], Nivedita Ravindran[1], Marie Beatrix John[1], Somaieh Hedayat[1], Holly E. Barker [1,9], James Wright[2], Lu Yu [2], Ioanna Mavrommati [1], Abigail Read[1], Barrie Peck [1,10], Mark Allen[3], Patrycja Gazinska[1], Helen N. Pemberton[1,4], Aditi Gulati[1,4], Sarah Nash[1], Farzana Noor[1], Naomi Guppy[1], Ioannis Roxanis[1], Guy Pratt[5], Ceri Oldreive [6], Tatjana Stankovic [6], Samantha Barlow[7], Helen Kalirai[7], Sarah E. Coupland[7], Ronan Broderick[2], Samar Alsafadi [8], Alexandre Houy [8], Marc-Henri Stern [8], Stephen Pettit [1,4], Jyoti S. Choudhary [2], Syed Haider [1], Wojciech Niedzwiedz [2], Christopher J. Lord [1,4] & Rachael Natrajan [1] ✉

*SF3B1* hotspot mutations are associated with a poor prognosis in several tumor types and lead to global disruption of canonical splicing. Through synthetic lethal drug screens, we identify that *SF3B1* mutant (*SF3B1*^MUT) cells are selectively sensitive to poly (ADP-ribose) polymerase inhibitors (PARPi), independent of hotspot mutation and tumor site. *SF3B1*^MUT cells display a defective response to PARPi-induced replication stress that occurs via downregulation of the cyclin-dependent kinase 2 interacting protein (CINP), leading to increased replication fork origin firing and loss of phosphorylated CHK1 (pCHK1; S317) induction. This results in subsequent failure to resolve DNA replication intermediates and $G_2/M$ cell cycle arrest. These defects are rescued through CINP overexpression, or further targeted by a combination of ataxia-telangiectasia mutated and PARP inhibition. In vivo, PARPi produce profound antitumor effects in multiple *SF3B1*^MUT cancer models and eliminate distant metastases. These data provide the rationale for testing the clinical efficacy of PARPi in a biomarker-driven, homologous recombination proficient, patient population.

Somatic mutations in components of the RNA splicing machinery occur across a variety of hematologic malignancies and solid tumors, highlighting the significance of aberrant splicing to tumorigenesis[1,2]. Heterozygous somatic hotspot mutations in the spliceosomal component *SF3B1* are the most common of these and occur at high frequencies in patients with myelodysplastic syndromes (20%), chronic lymphocytic leukemia (CLL; 15%), acute myeloid leukemia (AML; 3%) and in solid tumors such as uveal melanoma (20%), cutaneous melanoma (4%) and breast (2%), pancreatic (2%), lung (2%) and prostate cancer (1%)[3–13]. Hotspot *SF3B1* mutations are associated with poor patient outcomes in CLL, AML, uveal melanoma and breast cancer[14–18]. The *SF3B1* gene encodes subunit 1 of splicing factor 3b, a component of the U2 small nuclear ribonucleoprotein, which is involved in catalyzing precursor mRNA to mature transcripts. SF3B1 contains several HEAT domains (Huntingtin, Elongation factor 3, protein phosphatase 2A and Target of rapamycin 1), which are hotspots for most somatic mutations[1,2,8,19–22].

Hotspot mutations in *SF3B1* are neomorphic, inducing conformation changes in the HEAT superhelix domain that alters the interaction of SF3B1 with the pre-mRNA sequence[23]. As such, mutations result in reduced branchpoint fidelity, leading to the use of cryptic 3′ splice sites that lead to global aberrant splicing. Many of these transcripts are degraded via nonsense-mediated decay leading to the downregulation of mRNA and canonical proteins, while others produce aberrant proteins[1,8,19–22]. A large proportion of the alternative splicing events are conserved among multiple tumor types regardless of the mutated amino acid[21,24], and although these events have been comprehensively cataloged, their functional impact is largely uncharacterized.

*SF3B1* mutant (*SF3B1*^MUT) cells have been reported to rely on the wild-type allele for survival, while the heterozygous hotspot mutation leads to a neomorphic function, which does not produce a conventional oncogene addiction[25]. This suggests that therapeutic inhibition of the spliceosome may have a clinical benefit, particularly given many *SF3B1*^MUT cancers have few effective treatments[25]. We and others have demonstrated that *SF3B1*^MUT cancers are selectively sensitive to SF3b complex inhibitors both in vitro and in vivo[1,8,26,27], which has led to clinical efforts to directly inhibit the spliceosome in patients with refractory leukemia. However, preliminary clinical studies have shown minimal patient responses[28,29], suggesting other therapeutic approaches are warranted. Recent studies have identified aberrant splicing events that alter the maturation of the constitutive transcript and subsequent protein production of several genes. These lead to a failure in producing full-length proteins of a number of oncogenes and tumor suppressor genes, and consequently render *SF3B1*^MUT cells vulnerable to therapeutic intervention[30–33]. However, the clinical implementation of some of these approaches may be challenging.

## Results

### *SF3B1*^MUT cells show selective sensitivity to PARP inhibitors

To identify candidate therapeutic targets for cancers with *SF3B1* hotspot mutations, we utilized the leukemia K562^K700E (SF3B1^K700E) and parental (SF3B1^WT) isogenic cells[1], to model one of the most prevalent *SF3B1* hotspot mutations seen in patients[8,19,20] (Fig. 1a,b and Supplementary Fig. 1a). Using a drug-sensitivity screen, with an in-house curated library of 80 small-molecule inhibitors, we identified a series of candidate *SF3B1*^MUT synthetic lethal drugs, where at least two different concentrations significantly led to reduced survival in SF3B1^K700E cells[34] (survival fraction ratio K562^K700E/K562^WT cells < 0.6 and $P < 0.01$, unpaired two-tailed $t$-test; Fig. 1c and Supplementary Table 1). These included talazoparib (PARPi), gemcitabine, vinorelbine and SAR-20106 (CHK1 inhibitor; Fig. 1d and Supplementary Table 1). Subsequent validation in multiple isogenic cells with different hotspot mutations[19] identified a robust association with multiple PARPi, whereas additional hits from the screen failed to validate (Fig. 1e,f, Extended Data Fig. 1a–d and Supplementary Fig. 1b,c). PARPi sensitivity was also observed in the endogenously mutated uveal melanoma cell

line MEL202 harboring the most common uveal melanoma SF3B1^R625G hotspot variant[19] compared to a series of SF3B1^WT uveal melanoma cells (Fig. 1g and Extended Data Fig. 1e,g).

To confirm on-target effects, we used MEL202 SF3B1^R625G cells to knock in an inducible degron tag sequence (Degron-KI) into the single *SF3B1*^MUT allele as previously described[25]. In normal growth conditions, the mutant SF3B1 protein undergoes proteasomal degradation, and cells solely express the wild-type SF3B1 protein[25] (MEL202^R625G DD-SF3B1, hereafter termed MEL202^R625G-DEG (mutant degraded)). Exposure to the small-molecule ligand Shield-1 stabilized the degron-tagged mutant protein and reversed the aberrant splicing of the indicator transcript *CRNDE*. The continuous degradation of the SF3B1^MUT protein in these cells led to the loss of PARPi sensitivity, highlighting that mutant *SF3B1* influences PARPi response (Fig. 1g and Extended Data Fig. 1f).

We next used a genome-wide PARPi resistance (100 nM talazoparib) CRISPR knockout screen to gain mechanistic insights into the observed PARPi sensitivity in K562^K700E cells (Fig. 1h,i, Extended Data Fig. 1i,j and Supplementary Table 2). In agreement with previous studies[35,36], *PARP1* knockout led to PARPi resistance but had no significant effect on untreated cell viability (Fig. 1h,i, Extended Data Fig. 1k,l and Supplementary Table 2). Exposure to the PARP1 catalytic inhibitor veliparib showed limited sensitivity, compared to the more potent PARP-trapping agents, in *SF3B1*^MUT cells (Extended Data Figs. 1m and 2a,b). None of the previously identified genes, which were found to mediate PARPi resistance in homologous recombination-deficient *BRCA1*-defective cells[37], was significant in the knockout screen (Fig. 1h and Extended Data Fig. 2c). Consistent with this, *SF3B1*^MUT cells maintained their ability to form nuclear RAD51 foci at the sites of DNA damage, in contrast to the homologous recombination-deficient SUM149 *BRCA1*^MUT cells (Extended Data Fig. 2d), confirming that PARPi sensitivity in *SF3B1*^MUT cells is not driven by a possible deficiency in the homologous recombination machinery. Of note, there was no difference in SF3B1 protein expression between *SF3B1*^WT and *SF3B1*^MUT cells ± cycloheximide, suggesting that *SF3B1* hotspot mutations do not impact the protein expression or stability of SF3B1 (Extended Data Fig. 2e). Additionally, exposure of MEL202^R625G-DEG cells to the potent SF3B1 inhibitor Pladienolide B in combination with talazoparib did not sensitize MEL202^R625G-DEG cells to the same degree as single-agent PARPi exposure in MEL202^R625G cells. This agrees with existing data that *SF3B1* mutations are neomorphic rather than loss of function[1,25] (Extended Data Fig. 1f).

### *SF3B1*^MUT cells show dysregulation of ATR pathways

We sought to ascertain whether *SF3B1*^MUT cells showed changes in their repertoire of aberrant splicing events when exposed to PARPi. As previously described, K562^K700E cells had distinct transcriptomes, typified by unique changes to RNA splicing[1,8,18,19,21] (Extended Data Fig. 3a). PARPi exposure, however, resulted in only 17 significant differential splicing

---

**Fig. 1 | *SF3B1* hotspot mutations lead to PARPi sensitivity in isogenic models. a**, Lollipop plot of the number of *SF3B1* mutations in TCGA (pan-cancer cohort and MSK IMPACT clinical sequencing study (*n* = 21,912). Data from cBioportal. **b**, qRT–PCR of differentially spliced exons of selected indicator genes in the myeloid leukemia isogenic cell lines (K562) that express wild-type (WT) or mutant (K700E) *SF3B1* (*n* = 3 independent biological replicates). Data are mean ± s.e.m., unpaired two-tailed *t*-test; *CRNDE*, $P = 0.0003$; *ANKHD1*, $P = 0.0036$; *UQCC*, $P < 0.0001$ and *ABCC5*, $P < 0.0001$. **c**, Schematic of small-molecule inhibitor screening pipeline. **d**, Volcano plot of compound selectivity from the small-molecule inhibitor library screen in K562 cell lines ($-\log_{10} P < 0.01$ unpaired two-tailed *t*-test and surviving fraction (SF) ratio K562 SF3B1^K700E/SF3B1^WT < 0.6). Blue dots indicate two independent concentrations of the PARPi talazoparib. **e**, Fourteen-day clonogenic dose–response curves and representative images of K562 isogenic cells harboring the K700E SF3B1 hotspot variant and wild-type cells following exposure with the PARPi talazoparib (scale bar = 4 mm). **f**, Fourteen-day clonogenic dose–response curves of NALM-6

isogenic cells with the H662Q SF3B1 hotspot variant, K700K silent variant and wild-type cells following exposure with talazoparib and olaparib (*n* = 3 independent biological replicates, error bars show ± s.e.m.) **g**, Fourteen-day clonogenic dose–response curves of uveal melanoma MEL202^R625G cells with the endogenous R625G SF3B1 hotspot variant, and revertant MEL202^R625G-DEG cells following exposure with talazoparib. Data are mean normalized to DMSO control from *n* = 3 independent biological experiments, error bars show ± s.e.m (**e**–**g**). **h**, Waterfall plot of whole-genome CRISPR screen in K562 SF3B1^K700E cells, depicting hits (blue) from *n* = 3 independent biological replicate experiments. Genes known to cause resistance to PARPi in homologous recombination-deficient cells are highlighted. **i**, Bar plot depicting the SF_50 (concentration of drug that allows 50% cell survival) values of K562 *SF3B1* wild-type and K700E cells with Cas (control) or CRISPR *PARP1*^KO under talazoparib exposure (*n* = 3 independent biological repeats). Error bars show mean ± s.e.m. Unpaired two-tailed *t*-test, Cas9 wild-type versus K700E. *$P < 0.05$, **$P < 0.01$, ***$P < 0.001$, ****$P < 0.0001$ (**b**,**i**). SF, surviving fraction.

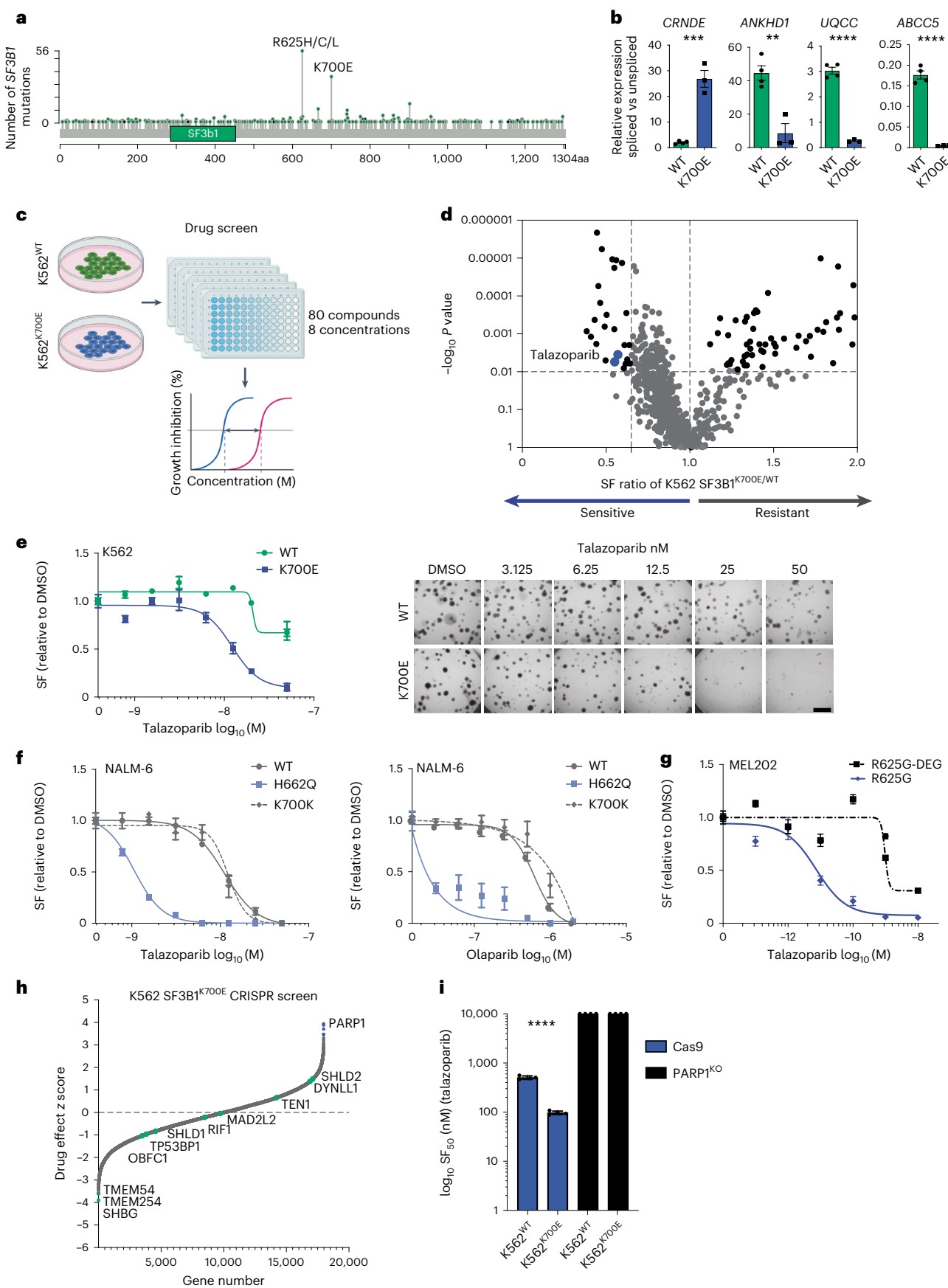

events and no changes in alternative splice site 3′ splice site recognition upon PARPi exposure (Extended Data Fig. 3a,b and Supplementary Tables 3 and 4), suggesting that PARPi exposure does not alter global splicing decisions in $SF3B1^{MUT}$ cells. Differential gene expression analysis similarly highlighted that PARPi induces minor transcriptional changes (Fig. 2a). Gene set enrichment analysis (GSEA) of the small number of differentially expressed genes identified that K562$^{K700E}$ cells showed specific dysregulation of genesets involved in transcription, DNA replication and the cell cycle compared to K562$^{WT}$ cells only when exposed to PARPi (Supplementary Fig. 2a,b and Supplementary Tables 3 and 4), suggesting that $SF3B1^{MUT}$ cells stop cycling and consequently alter their DNA replication and transcription upon PARPi exposure. Moreover, assessment of genome-wide RNA Pol II binding through ChIP–sequencing highlighted that $SF3B1^{MUT}$ cells do not have an innate transcriptional activity defect (that is no observed differential global RNA Pol II binding in untreated $SF3B1^{WT}$ versus $SF3B1^{MUT}$ cells), which could contribute to PARPi sensitivity in these cells (Extended Data Fig. 3c–e and Supplementary Table 5).

We then assessed what effects PARPi exposure had on the proteome of the MEL202$^{R625G}$ and isogenic MEL202$^{R625G-DEG}$ cells. As MEL202$^{R625G}$ cells possess a naturally occurring $SF3B1$ hotspot mutation; they have been shown to display the conserved mis-splicing signature associated with SF3B1$^{K700E}$ hotspot variations[1,19]; and were the most sensitive to PARPi, we reasoned that any differences in these cells would be marked further upon PARPi exposure (Supplementary Fig. 2c,d). Quantitative high-content peptide mass spectrometry ±PARPi identified that 54% of the proteome (4788/8856 identified proteins) was differentially expressed in MEL202$^{R625G}$ compared to MEL202$^{R625G-DEG}$ cells (Supplementary Table 6). GSEA analysis failed to identify any differentially enriched pathways between MEL202$^{R625G}$ and MEL202$^{R625G-DEG}$ cells exposed to DMSO; however, G$_2$/M checkpoint, apoptosis and E2F target genesets were selectively enriched after 48 h of 50 nM PARPi exposure in MEL202$^{R625G}$ cells (Fig. 2b,c and Extended Data Fig. 4a). The mass spectrometry data additionally identified several ataxia-telangiectasia mutated and Rad3-related (ATR) pathway-related proteins as significantly downregulated in MEL202$^{R625G}$ compared to MEL202$^{R625G-DEG}$ cells (log$_2$-transformed fold change < −2), including DYRK2, RAD9A, CINP, TTI1, TTI2 and NEK1. Of these, CINP was further downregulated upon PARPi exposure and was the most downregulated protein in MEL202$^{R625G}$ cells compared with MEL202$^{R625G-DEG}$ cells exposed to PARPi (Fig. 2b and Supplementary Table 6). CINP is associated with genome maintenance and found to transiently interact with ATRIP-ATR, although not specifically, under UV-induced DNA damage[38]. CINP protein expression was downregulated in multiple $SF3B1^{MUT}$ cells and patient-derived uveal melanoma models compared to $SF3B1^{WT}$ models (Fig. 2d–f, Extended Data Fig. 4b and Supplementary Fig. 2e). This association was also validated in primary SF3B1$^{K700E}$ patients, who were treated with single-agent olaparib as part of the dose-finding phase 1 PiCCLe clinical trial[39]. Three of the four $SF3B1^{MUT}$ patients had the longest progression-free survival time on olaparib and showed loss of CINP protein expression (Fig. 2g,h).

Analysis of mis-spliced events that were identified in $SF3B1^{MUT}$ primary cancers harboring multiple hotspot mutations from published studies[2,15], that were also identified in the MEL202$^{R625G}$ cells, failed to identify any mRNA downregulation or aberrant splicing event of CINP, which may explain the observed reduction in protein levels (Supplementary Fig. 2f and Supplementary Table 4). We also did not identify any significant alternative splicing event of additional genes directly involved in the ATR pathway (Supplementary Table 7). Moreover, MEL202$^{R625G-DEG}$ and MEL202$^{R625G}$ cells expressed similar levels of ATRIP (immediate interactor of CINP), following short-term DMSO or PARPi exposure (Extended Data Fig. 4c). Additionally, we observed no stabilization of CINP protein expression in MEL202$^{R625G}$ cells upon inhibition of nonsense-mediated decay (cycloheximide) or proteasome inhibition (MG-132; Extended Data Fig. 4d).

### $SF3B1^{MUT}$ cells show a defective replication stress response

Given the observed G$_2$/M checkpoint induction in MEL202$^{R625G}$ cells and the downregulation of proteins involved in the ATR-mediated replication stress response, we evaluated whether $SF3B1^{MUT}$ cells have defects in replication stress[40,41]. Using DNA-fiber assays to assess DNA replication origin firing, fork speed and symmetry[42,43], we observed no difference in the number of origins, fork speed or sister fork ratio under normal growth conditions. However, (3 h) PARPi exposure in MEL202$^{R625G}$ cells resulted in a sustained number of newly firing replication origins (as verified upon CDC7 inhibition), a significant increase in fork speed and an increase in the sister fork ratio (that is, fork asymmetry) compared to MEL202$^{R625G-DEG}$ cells (Fig. 3a–d and Extended Data Fig. 5a,b). As such, (1–3 h) PARPi exposure resulted in reduced induction of pCHK1 (S317) and pATR (T1989) in MEL202$^{R625G}$ and K700E$^{K700E}$ cells, whereas MEL202$^{R625G-DEG}$, MP41$^{WT}$ and K562$^{WT}$ cells showed a time-dependent induction of the replication stress response. This was coupled with a decrease in pRPA2 (S33) and an increase in total RPA foci in $SF3B1^{MUT}$ cells, highlighting an increase in DNA/RPA complexes due to perturbed replication (replication stress following PARPi exposure; Fig. 3e,f and Extended Data Fig. 5c,d).

$CINP$ gene silencing similarly resulted in an impaired pCHK1 (S317) response and caused sensitivity to PARPi in MEL202$^{R625G-DEG}$ cells (Fig. 4a,b and Extended Data Fig. 5e). Of note, $CINP$ gene silencing did not significantly further the sensitivity of MEL202$^{R625G}$ cells to talazoparib. Hydroxyurea, known to collapse replication forks, did not reproduce this defective response, as fork symmetry (CIdU/IdU) and pCHK1 (S317) induction were comparable between MEL202$^{R625G-DEG}$ and MEL202$^{R625G}$ cells. Furthermore, cell survival after hydroxyurea or gemcitabine addition showed no selectivity for MEL202$^{R625G}$ cells, indicating that PARPi sensitivity in $SF3B1^{MUT}$ cells is driven by a defective replication stress response to an increase in fork origin firing and subsequent accelerated replication, rather than innate replication stress (Extended Data Figs. 5f and 6a–c). Reconstitution of CINP protein expression in MEL202$^{R625G}$ cells (MEL202$^{R625G}$–CINP–GFP) resulted in restoration of the canonical replication stress response to PARPi (pCHK1 (S317) induction and reversal of PARPi sensitivity),

---

**Fig. 2 | $SF3B1^{MUT}$ cells show transcriptional dysregulation and the induction of G$_2$/M checkpoint proteins when exposed to PARPi. a**, MA plots highlighting the significantly differentially expressed genes between the highlighted comparisons in the K562 RNA-sequencing data (DMSO K562$^{K700E}$ versus DMSO K562$^{WT}$ changes just due to the $SF3B1$ mutation and PARPi K562$^{K700E}$ versus PARPi K562$^{WT}$ interaction; changes due to the effect of PARPi only accounting for the genotype-specific effects). Significantly differentially expressed genes are depicted in red (FDR < 0.01, |LFC| > 1). **b**, Heatmap representing mean-centered, hierarchical clustering of proteins and samples mapping to the ATR pathway from the total-MS/MS. **c**, Gene set enrichment plot from GSEA analysis of total-MS/MS of MEL202$^{R625G}$ and MEL202$^{R625G-DEG}$ isogenic cell lines after DMSO or 50 nM talazoparib exposure for 48 h. $P$ values shown are FDR corrected.

**d**, Representative micrographs of CINP IHC in $SF3B1^{MUT}$ and $SF3B1^{WT}$ PDX models. Scale bar, 200 μm. **e**, Box and whiskers plot of the digital quantification of CINP IHC across $SF3B1^{MUT}$ ($n = 3$) and $SF3B1^{WT}$ ($n = 8$) PDX models ($P = 0.0519$, Welch's unpaired two-tailed $t$-test). **f**, Western blot of CINP from $SF3B1^{MUT}$ and $SF3B1^{WT}$ PDX lysates and β-actin loading control. **g**, Heatmap depicting the distribution of genetic alterations in CLL driver genes: $ATM$, $SF3B1$ and $TP53$ aligned according to time on olaparib treatment. Presence of mutations is highlighted by green shaded boxes. Modified from ref. 39. **h**, Western blot of CINP expression in SF3B1$^{WT}$ and SF3B1$^{K700E}$ patients enrolled in the PiCCLe trial collected at baseline and exposed to PARPi for 48 h in vitro before lysis and western blot analysis. $P$ values shown are calculated with chi-square test (**d**). IHC, immunohistochemistry.

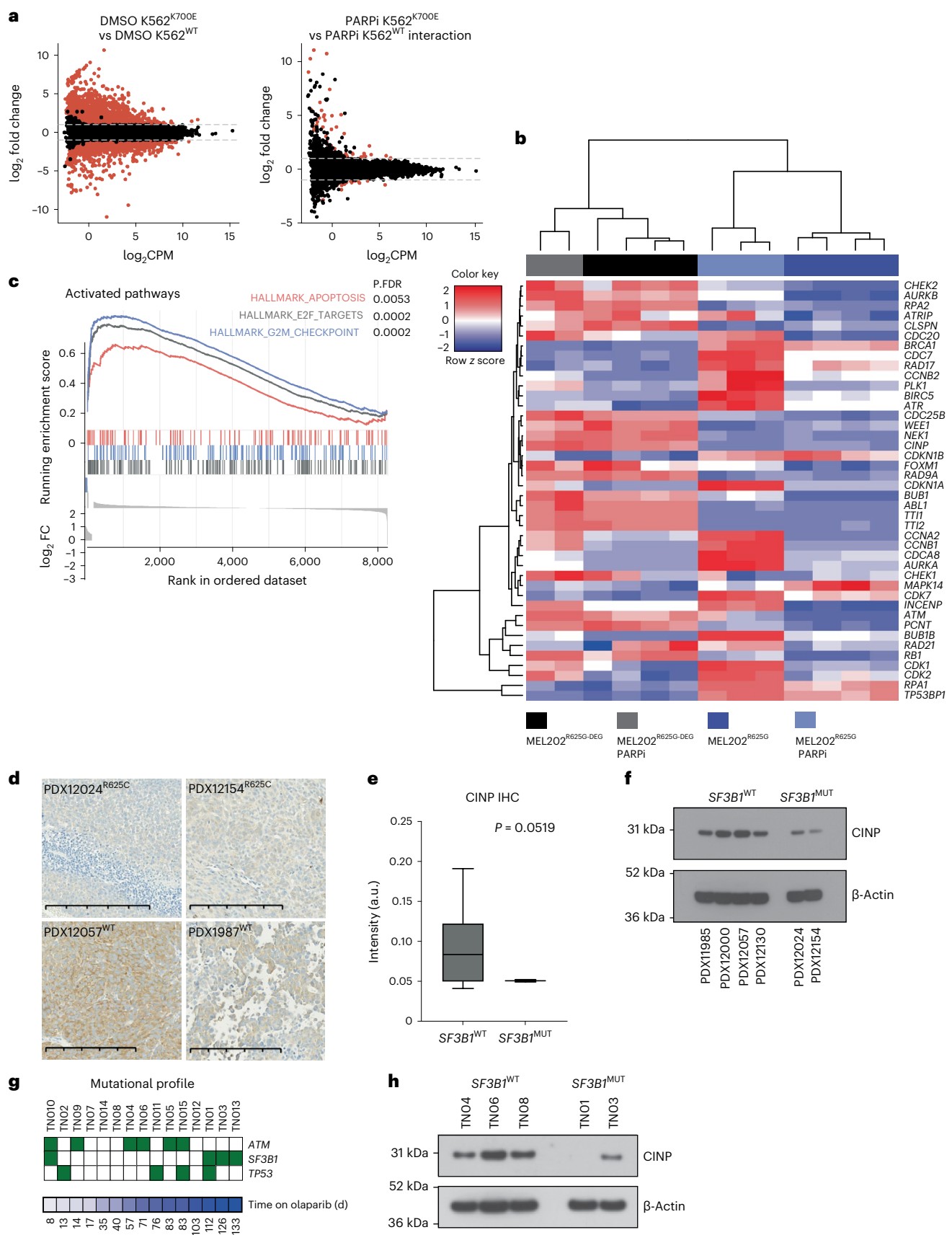

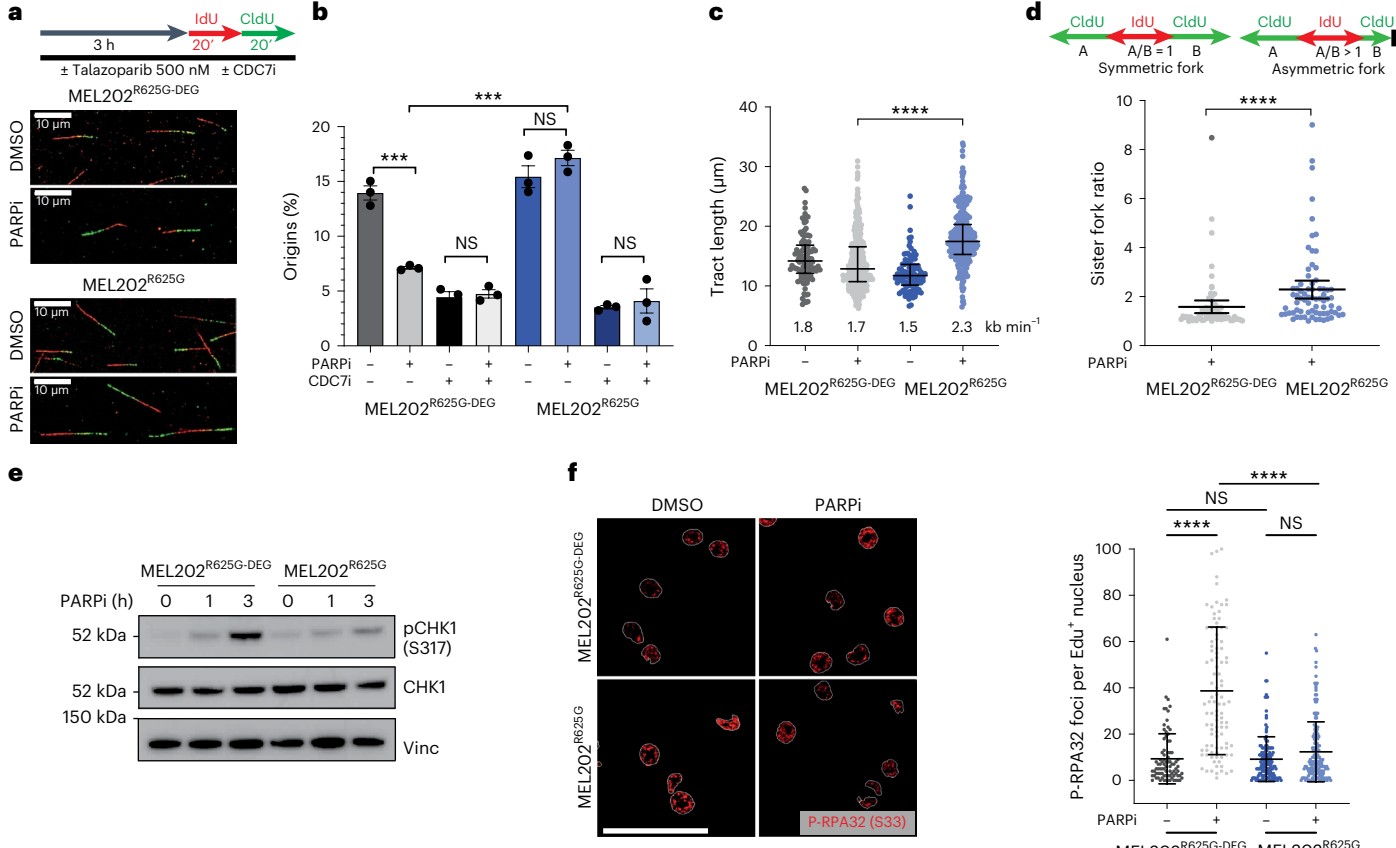

**Fig. 3 | *SF3B1*<sup>MUT</sup> cells elicit a defective replication stress response following PARPi exposure. a**, Experimental setup of fiber assay. Cells were pre-incubated with 500 nM talazoparib for 3 h, followed by sequential labeling with 25 µM IdU (red) and 125 µM CldU (green). Representative immunofluorescence images of individual fibers highlighting the differences in tract length. **b**, Bar plot showing percentage of newly firing origins from IdU and CldU labeled DNA fibers after 3 h 500 nM talazoparib or DMSO or combination 500 nM talazoparib and 20 µM CDC7i XL413. A minimum of 400 replication structures were scored across *n* = 3 biologically independent experiments and the percentage of origins was calculated in each of the replicate experiments. ***P = 0.0005 and ***P = 0.0002 (left to right), unpaired two-tailed *t*-test. **c**, Scatterplot of fork speed (tract length). **d**, Schematic of scoring and scatterplot of sister fork ratio. Fork symmetry was analyzed by calculating the ratio of the leftward and rightward tracts emanating by sister forks emerging from the same replication origin; A/B ratio >1 indicates fork asymmetry and increased fork stalling. Data are mean of *n* = 3 biological replicates, error bars show ±s.e.m. *P* value determined by unpaired two-tailed *t*-test. **e**, Western blot of CHK1 phosphorylation at serine 317 (pCHK1 (S317)), and total CHK1 expression in MEL202 isogenic cells, after 0, 1 and 3 h of 500 nM talazoparib exposure. Images are representative of *n* = 3 biological replicates. **f**, Representative immunofluorescence images (left) and scatterplot (right) of pRPA32 (S33) foci in MEL202 isogenic cells following 3 h of 500 nM talazoparib or DMSO exposure. Data are from *n* = 2 biological replicates, error bars show ± s.d. of foci in individual nuclei. Scale bar, 50 µm *P* values determined by unpaired two-tailed *t*-test. ***P < 0.001, ****P < 0.0001. NS, not significant.

validating that the defective replication stress response is directly due to low levels of CINP in *SF3B1*<sup>MUT</sup> cells (Fig. 4c,d and Supplementary Fig. 4a).

We next investigated the consequence of the *SF3B1*<sup>MUT</sup>-specific replication stress response. MEL202<sup>R625G-DEG</sup> and K562<sup>WT</sup> cells displayed robust recruitment of 53BP1 and γH2AX foci after 3 h PARPi exposure, coinciding with pCHK1 (S317) activation. MEL202<sup>R625G</sup> and K562<sup>K700E</sup> cells failed to recruit 53BP1 upon 3 h PARPi exposure, paralleling their lack of pCHK1 (S317) induction, although showed γH2AX induction, indicative of the duality of 53BP1. However, after 48 h PARPi exposure, the majority of MEL202<sup>R625G</sup> and K562<sup>K700E</sup> cells induced 53BP1 and showed sustained γH2AX foci, whereas *SF3B1*<sup>WT</sup> cells resolved these foci (Fig. 4e,f and Extended Data Fig. 7a,b). These phenotypes were reversed upon CINP overexpression (Extended Data Fig. 7c,d) and in accordance with the pCHK1 (S317) response were not recapitulated under hydroxyurea exposure (Extended Data Fig. 7e,f).

We then sought to address whether the defective replication stress response observed in *SF3B1*<sup>MUT</sup> cells persists due to incomplete fork repair and replication. Using FANCD2 foci formation as a marker of

unresolved replication intermediates, we observed no significant increase in FANCD2 foci in MEL202<sup>R625G-DEG</sup> cells exposed to PARPi compared to DMSO controls. This was coupled with an increase in the percentage of MUS81-positive FANCD2 foci, indicative of the resolution of replication intermediates[44,45] (Fig. 4g,h and Extended Data Fig. 8a). MEL202<sup>R625G</sup> cells, however, showed a significant increase in the number of FANCD2 foci after PARPi exposure and a reduction of MUS81-positive FANCD2 foci. This is suggestive of impaired recruitment to damaged forks, which results in incomplete replication and unresolved replication intermediates in *SF3B1*<sup>MUT</sup> cells. Accordingly, siRNA-mediated silencing of *MUS81* in MEL202<sup>R625G</sup> cells induced no further sensitivity to PARPi, in contrast to the observed interaction of *MUS81* silencing in *BRCA* mutant cells[46] (Extended Data Fig. 8b,c). These markers of unresolved fork structures were observed under the same PARPi concentration and exposure time as the total mass spectrometry dataset, indicating that after failing to activate a canonical replication stress response to PARPi, *SF3B1*<sup>MUT</sup> cells express proteins integral to the G<sub>2</sub>/M checkpoint.

Rescue of the defective replication stress response was additionally validated through the generation of an independent

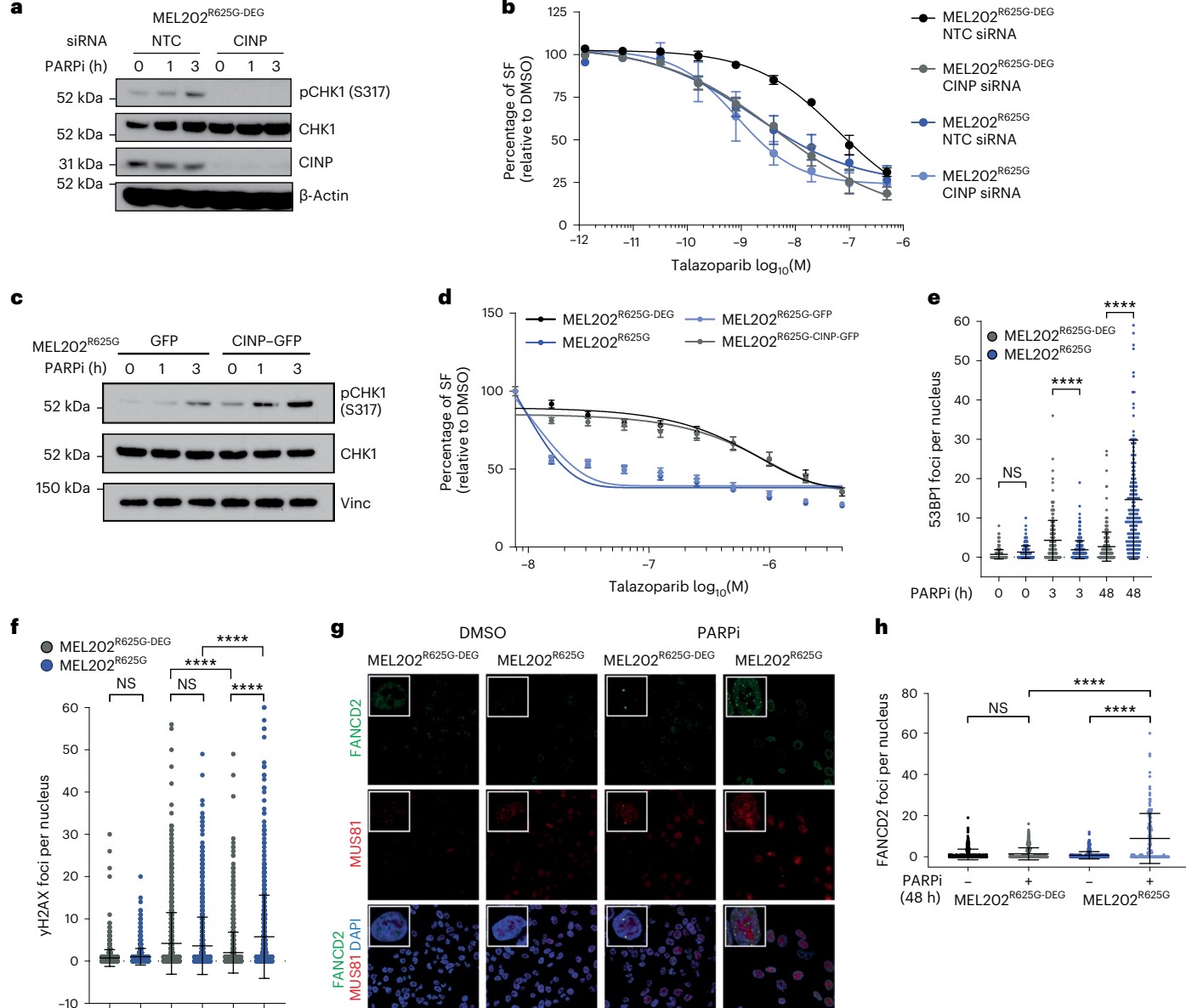

**Fig. 4 | A defective replication stress response leads to PARPi sensitivity in *SF3B1*[MUT] cells. a,** Western blot of pCHK1 (S317), total CHK1 and CINP expression in MEL202[R625G-DEG] cells after non-targeting control (NTC) or *CINP* siRNA-mediated gene silencing, with 0, 1 or 3 h of 500 nM talazoparib exposure. Images are representative of *n* = 3 biological replicates. **b,** Talazoparib dose–response curves showing the SF, relative to DMSO, of MEL202 isogenic cells after NTC or *CINP* siRNA-mediated gene silencing. Data are mean of three replicates, error bars show ±s.e.m. **c,** Western blot of pCHK1 (S317) and total CHK1 expression in MEL202[R625G] cells expressing control–GFP or CINP–GFP, following 0, 1 and 3 h of 500 nM talazoparib exposure. Images are representative of two biological replicates. **d,** Talazoparib dose–response curves showing the SF, relative to DMSO, of MEL202 isogenic cells, and MEL202[R625G] cells expressing control–GFP or CINP–GFP. Data are mean of *n* = 3 biological replicates, error bars show ±s.e.m. **e,f,** Scatterplots showing the number of 53BP1 (**e**) and γH2AX (**f**) foci per nucleus in MEL202 isogenic cells after 0, 3 h (500 nM) and 48 h (50 nM) talazoparib exposure. Data are representative of *n* = 3 biological replicates, error bars show ±s.d. **g,h,** Representative immunofluorescence images (**g**) of FANCD2 and MUS81 foci and scatterplot of FANCD2 foci (**h**) in MEL202 isogenic cells after 48 h DMSO or (50 nM) talazoparib exposure. Scale bar, 100 μm. Data are representative of *n* = 3 biological replicates, error bars show ±s.d. of individual nuclei assessed. *P* values are calculated by one-way ANOVA (**e**, **f** and **h**), ****$P < 0.0001$. NTC, nontargeting control; NS, not significant.

CINP overexpressing cell line model (Extended Data Fig. 8d–i). MEL202[R625G] and K562[K700E] cells were not selectively sensitive to single-agent ATR or CHK1 inhibition, suggesting that replication-induced R loops resulting in ATR activation are not a primary mechanism of sensitivity in *SF3B1*[MUT] cells[47,48] (Supplementary Figs. 1c and 4b). Together, these results indicate that PARPi sensitivity in *SF3B1*[MUT] cells is driven by a defective replication stress response to increased fork origin firing and incomplete resolution of replication intermediates.

## *SF3B1*[MUT] cells stall in G2/M upon PARPi exposure

Finding that *SF3B1*[MUT] cells harbor markers of unresolved replication intermediates upon PARPi exposure, and that temporally this coincides with the induction of G2/M checkpoint proteins in the mass spectrometry analysis, we performed a cell cycle analysis ± PARPi. Propidium iodide staining and fluorescence-activated cell sorting (FACS) analysis showed a similar cell cycle profile in MEL202[R625G-DEG] and MEL202[R625G] cells. Upon 48 h PARPi exposure, MEL202[R625G] cells showed a significant increase in the percentage of cells in G2/M, which

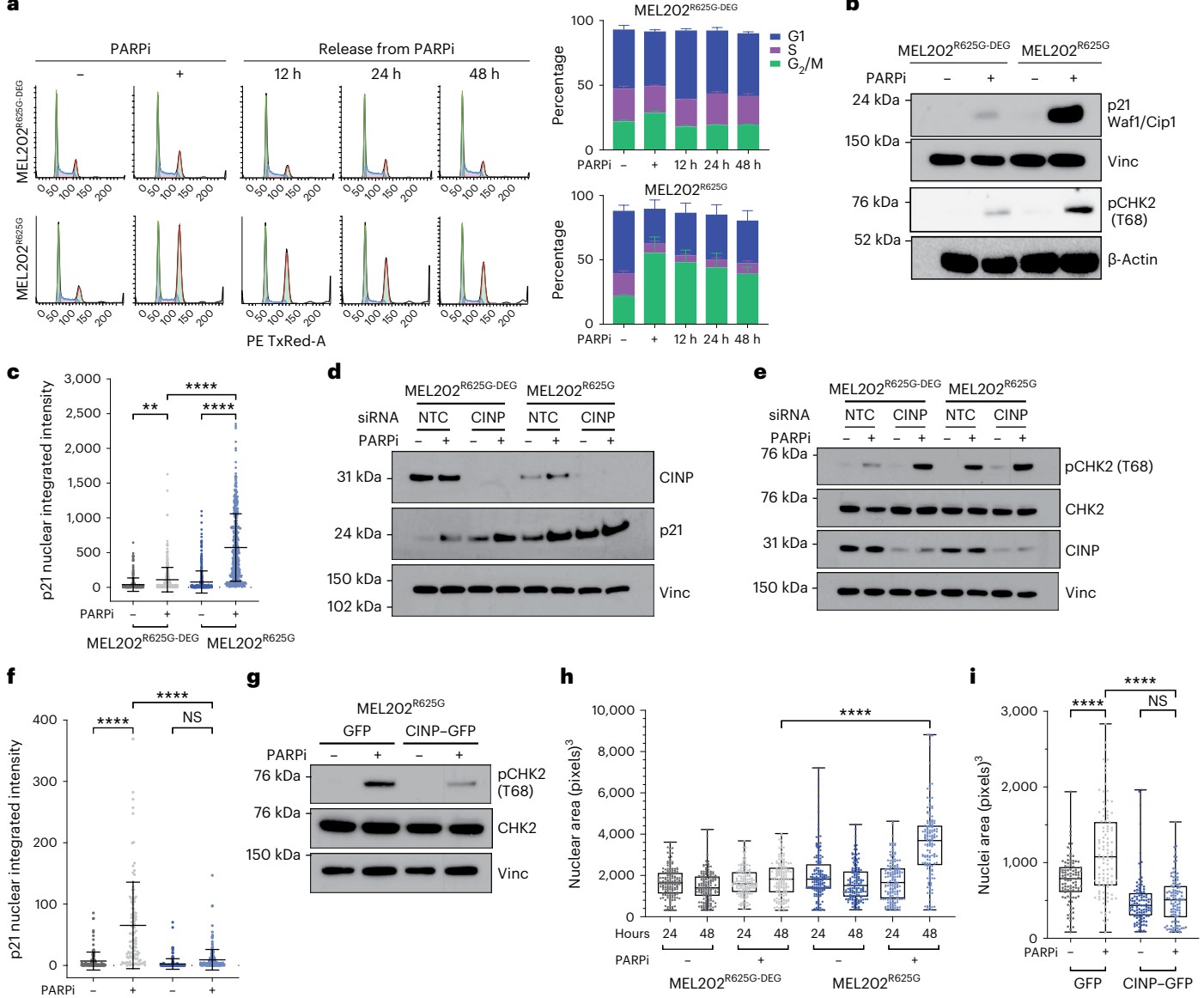

**Fig. 5 | PARP inhibition leads to G₂/M checkpoint stalling in *SF3B1*ᴹᵁᵀ cells.**
**a**, Flow cytometry histograms of propidium iodide staining and stacked bar plots, showing the cell cycle profile of MEL202 isogenic cells after 48 h of 50 nM talazoparib exposure, and 12, 24 and 48 h after subsequent talazoparib removal. Data are mean of *n* = 3 biological replicates, error bars show ±s.e.m. **b**, Western blot of p21ᵂᵃᶠ¹/Cip1 and CHK2 phosphorylation (threonine68 (pCHK2 (T68)) in MEL202 isogenic cells after 48 h of 50 nM talazoparib or DMSO exposure. Images are representative of *n* = 3 biological replicates. **c**, Scatterplot quantification of nuclear intensity of p21 in MEL202 isogenic cells after 48 h of 50 nM talazoparib or DMSO exposure. Data representative of *n* = 4 biological replicates, error bars show ±s.d. \*\**P* = 0.0021, \*\*\*\**P* < 0.0001, one-way ANOVA. **d**, Western blot showing expression of CINP and p21 in MEL202 isogenic cells after NTC or *CINP* gene silencing 48 h after 50 nM talazoparib exposure. Data are representative of *n* = 2 biological replicates. **e**, Western blot of pCHK2 (T68) and CINP expression in MEL202 isogenic cells after NTC or *CINP* gene silencing and 48 h of 50 nM

talazoparib or DMSO exposure. Images are representative of *n* = 3 biological replicates. **f**, Scatterplot showing the nuclear intensity of p21 in MEL202ᴿ⁶²⁵ᴳ cells expressing control–GFP or CINP–GFP, after 48 h of 50 nM talazoparib or DMSO exposure. Data are representative of *n* = 2 biological replicates, error bars show ±s.d. \*\*\*\**P* < 0.0001, one-way ANOVA. **g**, Western blot of pCHK2 (T68) and total CHK2 expression in MEL202ᴿ⁶²⁵ᴳ cells expressing control–GFP or CINP–GFP, after 48 h of 50 nM talazoparib or DMSO exposure. Images are representative of *n* = 2 biological replicates. **h**, Box and whiskers plot showing nuclear area of MEL202 isogenic cells after 24 h and 48 h of 50 nM talazoparib or DMSO exposure. Data are mean of three biological replicates, error bars show ±s.e.m. **i**, Box and whiskers plot depicting nuclear area of MEL202ᴿ⁶²⁵ᴳ cells expressing control–GFP or CINP–GFP, after 48 h of 50 nM talazoparib or DMSO exposure. Data are mean of *n* = 3 biological replicates, error bars show minimum to maximum nuclear area. \*\*\*\**P* < 0.0001, one-way ANOVA (**h** and **i**).

was predominately maintained after drug wash off, whereas the MEL202ᴿ⁶²⁵ᴳ⁻ᴰᴱᴳ cells showed no significant response to PARPi exposure (Fig. 5a and Extended Data Fig. 9a). Further investigation quantified that the immediate accumulation of cells in G₂/M corresponds with a decrease in the G₁ population, rather than a difference in the

proportion of cells in S phase following PARPi exposure (Extended Data Fig. 9b). PARPi exposure produced a dose-dependent induction of pCHK1 (S345) in MEL202ᴿ⁶²⁵ᴳ cells at 48 h, indicating a functional ATR/CHK1 DNA damage response, coinciding with the recruitment of 53BP1 and γH2AX to sites of DNA damage (Extended Data Fig. 9c and

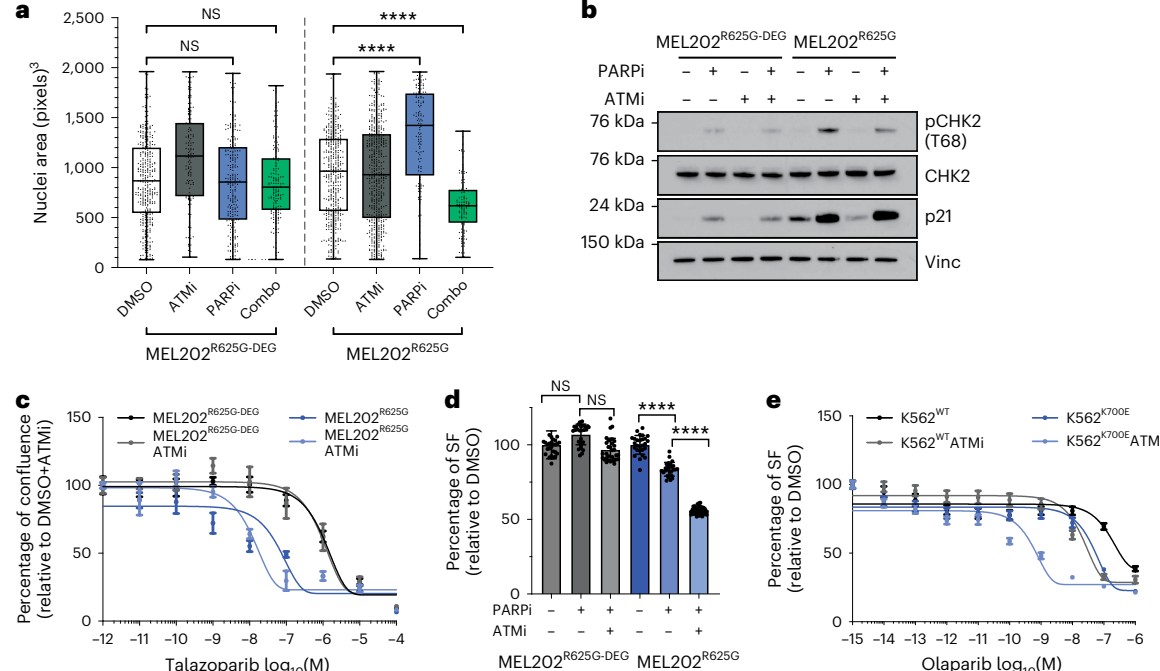

**Fig. 6 | PARPi and ATMi combination treatment lower the G$_2$/M checkpoint.**
**a,b**, Box and whiskers plot showing nuclear area (**a**), and western blot of pCHK2 (T68), total CHK2 and p21 expression in MEL202 isogenic cells after 48 h of exposure with 400 nM ATMi (KU-55933), 50 nM talazoparib, combination exposure or DMSO (**b**). $n = 3$ independent biological replicates, error bars show minimum to maximum nuclear area. ****$P < 0.0001$, one-way ANOVA. **c**, Talazoparib dose–response curves of MEL202 isogenic cells treated with DMSO or ATMi KU-55933. Data are mean of $n = 3$ biological replicates, error bars show ±s.e.m. **d**, Column bar graph showing the relative survival of MEL202 isogenic cells after days of exposure with 50 nM talazoparib, combination with ATMi AZD0156 or DMSO. Data are mean of $n = 3$ biological replicates, error bars show ±s.e.m. **e**, Dose–response curves of K562 isogenic cells exposed to olaparib in combination with DMSO or ATMi AZD0156. Data are mean of $n = 3$ replicates, error bars show ±s.e.m. ****$P < 0.0001$, unpaired two-tailed $t$-test.

Fig. 4e,f). This paralleled with increased pATM (S1981) in MEL202$^{R625G}$ cells, which canonically induced pCHK2 (T68) and nuclear p21 (Waf/Cip1) protein expression in both MEL202$^{R625G}$ and K562$^{K700E}$ cells (Fig. 2b, Fig. 5b,c and Extended Data Fig. 9d–f). This activity is reported to inhibit the kinase activity of CDK1-cyclin B, thus blocking progression through G$_2$/M[49]. Consistent with our earlier observations, these phenotypes were also observed in MEL202$^{R625G-DEG}$ cells upon *CINP* gene silencing (Fig. 5d,e, Extended Data Fig. 9g,h and Supplementary Fig. 5a) and could be rescued through the reexpression of CINP in MEL202$^{R625G}$ cells (Fig. 5f,g and Supplementary Fig. 5b). Of note, PARPi exposure here did not lead to high levels of single strand or double strand DNA damage measurable by alkaline and neutral COMET assays, respectively, in either MEL202$^{R625G-DEG}$ or MEL202$^{R625G}$ cells (Supplementary Fig. 5c–g). This is an outcome in agreement with studies highlighting that higher levels of DSBs only occur upon progression into a subsequent cell cycle after PARPi exposure[42,50].

Using BIRC5 (survivin), which acts as a subunit of the chromosomal passenger complex (CPC) to regulate key mitotic events[51,52], to assess the mitotic phases under PARPi exposure, we observed normal mitotic progression of MEL202$^{R625G-DEG}$ cells, whereas MEL202$^{R625G}$ cells were entirely in interphase. MEL202$^{R625G}$ cells showed nuclear translocation of survivin (Supplementary Fig. 5h,i), supporting the notion that survivin has a dual role as an apoptosis inhibitor and a mitotic effector, where a change from antiapoptotic to CPC function occurs in G$_2$/M as the cells prepare for mitosis[51,52]. We also observed a significant reduction in the percentage of mitoses (using the marker phospho-histone H3 (S10)[53]) in MEL202$^{R625G}$ cells exposed to PARPi (Supplementary Fig. 5j), which was accompanied by an increase in nuclear area and pericentrin area (Fig. 5h and Supplementary Fig. 6a,b), indicative of cells at the G$_2$/M checkpoint[4]. This was also rescued in MEL202$^{R625G}$ cells overexpressing CINP (Fig. 5i and Extended Data Fig. 8d–h).

These results indicate that the deficient response of *SF3B1*$^{MUT}$ cells to the replication stress caused by PARPi exposure leads to increased fork origin firing, a subsequent increase in unresolved replication intermediates and activation of the G$_2$/M checkpoint. By reexpressing CINP in *SF3B1*$^{MUT}$ cells, we can rescue the DNA damage and G$_2$/M checkpoint activation caused by PARP inhibition, and ultimately, the sensitivity of *SF3B1*$^{MUT}$ cells to talazoparib (Fig. 4d and Extended Data Fig. 8i).

Given our observations that G$_2$/M checkpoint activation upon PARPi exposure is primarily regulated by the ataxia-telangiectasia mutated (ATM)/CHK2 pathway in *SF3B1*$^{MUT}$ cells[54], we hypothesized that treating *SF3B1*$^{MUT}$ cells with combinations of PARPi and ATM inhibitors (ATMi) would abrogate G$_2$/M stalling, leading to further cell death. In contrast to the increase in nuclear area caused by single-agent PARPi, a combination of talazoparib with the ATMi KU-55933 resulted in a significant reduction in nuclear area in MEL202$^{R625G}$ cells compared to PARPi or DMSO (Fig. 6a). Consistent with this, we observed a reduction in pCHK2 (T68) phosphorylation with the combination of PARPi with either KU-55933 (Fig. 6b) or the more potent ATMi AZD0156 (Supplementary Fig. 6c). Finally, PARPi and ATMi combinations led to a significant reduction in the viability of both MEL202$^{R625G}$ and K562$^{K700E}$ cells compared to PARPi alone (Fig. 6c–e and Supplementary Fig. 6d). In contrast to this, the combination of either CHK1i or ATRi with talazoparib was not selective in MEL202$^{R625G}$ cells (Supplementary Fig. 6e,f).

## PARPi suppress *SF3B1*$^{MUT}$ growth and metastasis in vivo

We subsequently tested the in vivo therapeutic potential of single-agent PARPi talazoparib in *SF3B1*$^{MUT}$ cells. In both, MEL202$^{R625G-DEG}$ and MEL202$^{R625G}$ tumor-bearing mice that received the drug vehicle alone, tumor growth continued unabated and liver micrometastases were observed in all mice (Fig. 7a–d). In contrast, talazoparib treatment (0.33 mg kg$^{-1}$) had a strong antitumor effect in the MEL202$^{R625G}$

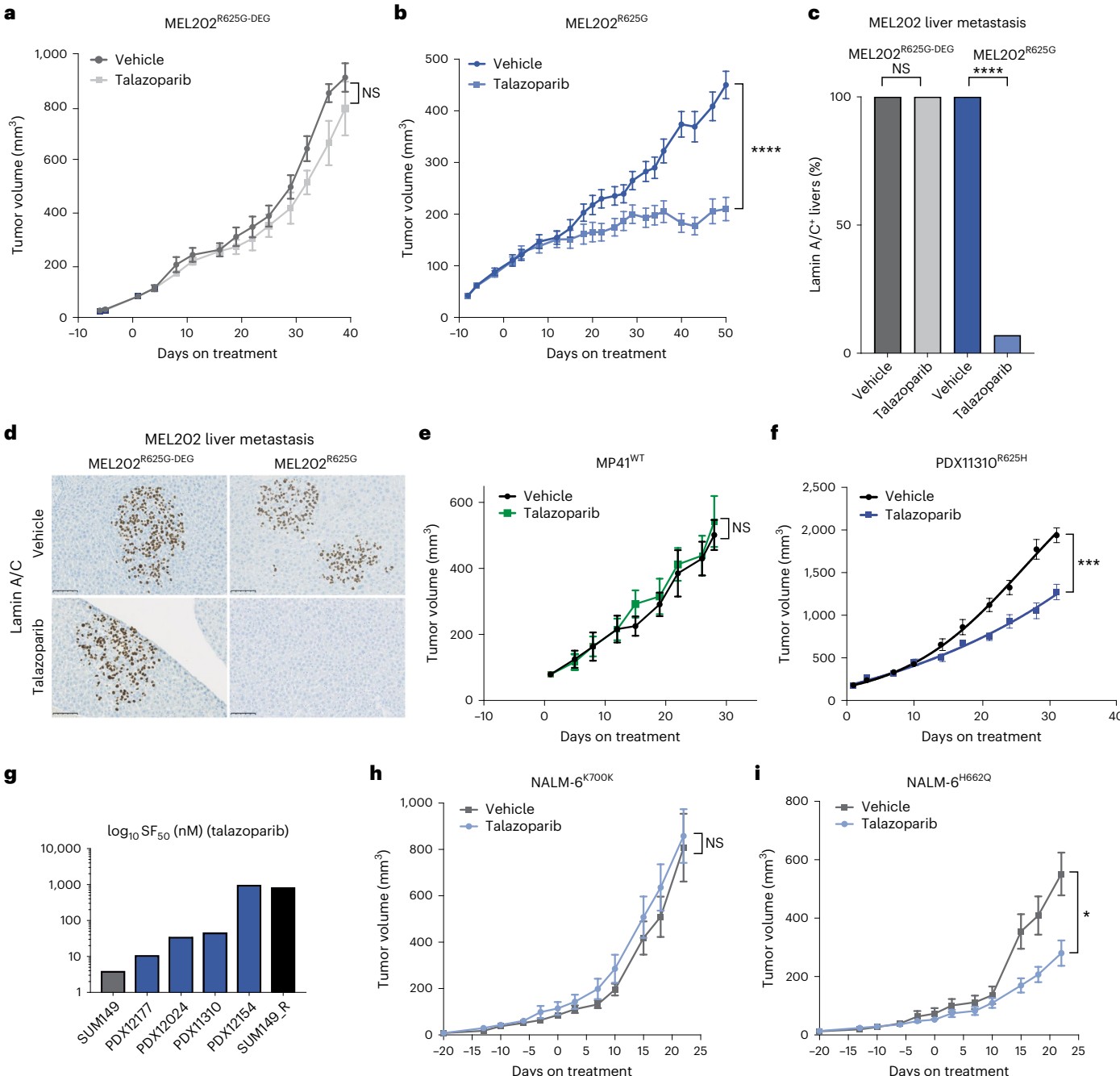

**Fig. 7 | PARP inhibition suppresses _SF3B1_[MUT] tumor growth and metastasis in vivo. a**, Chart depicting tumor volume of the therapeutic response to talazoparib treatment in NSG-nude mice bearing MEL202[R625G-DEG] xenograft tumors over time (0.33 mg kg[−1]). Day 0 represents the first day of treatment. Tumors, vehicle $n = 8$, talazoparib $n = 9$. NS, $P = 0.1825$, two-way ANOVA. **b**, Chart depicting tumor volume of the therapeutic response to talazoparib treatment in NSG-nude mice bearing _SF3B1_[MUT] MEL202[R625G] xenograft tumors over time, (0.33 mg kg[−1]). Day 0 represents the first day of treatment. Tumors, vehicle $n = 16$, talazoparib $n = 15$. ****$P < 0.0001$, two-way ANOVA. **c**, Bar plot of number of mice with or without human lamin A/C positive cells in liver sections, representing liver metastasis of all MEL202[R625G-DEG] and MEL202[R625G] subcutaneous tumors under talazoparib treatment. ****$P < 0.0001$, unpaired two-tailed _t_-test. **d**, Representative images of immunohistochemical assay of mouse livers from the MEL202[R625G-DEG] and MEL202[R625G] cells grown in vivo. Scale bar, 100 μm. **e,f**, Chart depicting tumor volume of the therapeutic response to talazoparib treatment in NSG-nude mice

bearing _SF3B1_[WT] and _SF3B1_[MUT] patient-derived xenograft tumors MP41[WT] (**e**) and PDX11310[R625H] (**f**) over time, (0.33 mg kg[−1]). Day 0 represents the first day of treatment. MP41[WT] tumors, vehicle $n = 9$, talazoparib $n = 9$. NS, $P = 0.6536$, two-way ANOVA. PDX11310[R625H] tumors, vehicle $n = 10$, talazoparib $n = 10$. ***$P = 0.0005$, two-way ANOVA. **g**, Bar plot of SF[50] concentrations of talazoparib efficacy in a series of _SF3B1_[MUT] patient-derived organoids (R625C (PDX12177, PDX12024 and PDX12154), R625H (PDX11310) grown ex vivo. Three-dimensional cultures of the _BRCA1_[MUT] SUM149 and revertant SUM149 cell lines were used as controls of PARPi sensitivity, respectively. Data are mean of $n = 1$ biological replicate and $n = 6$ technical replicates, error bars show ±s.e.m. **h,i**, Growth charts depicting tumor volume of the therapeutic response to talazoparib treatment of NALM-6[K700K] _SF3B1_[WT] tumors and NALM-6[H662Q] _SF3B1_[MUT] tumors over time in CB-17 mice (0.33 mg kg[−1]). NALM-6[K700K] _SF3B1_[WT] tumors, vehicle $n = 6$, talazoparib $n = 8$. NS, $P = 0.4356$, two-way ANOVA. NALM-6[H662Q] _SF3B1_[MUT] tumors, vehicle $n = 8$, talazoparib $n = 8$. *$P = 0.0388$, two-way ANOVA.

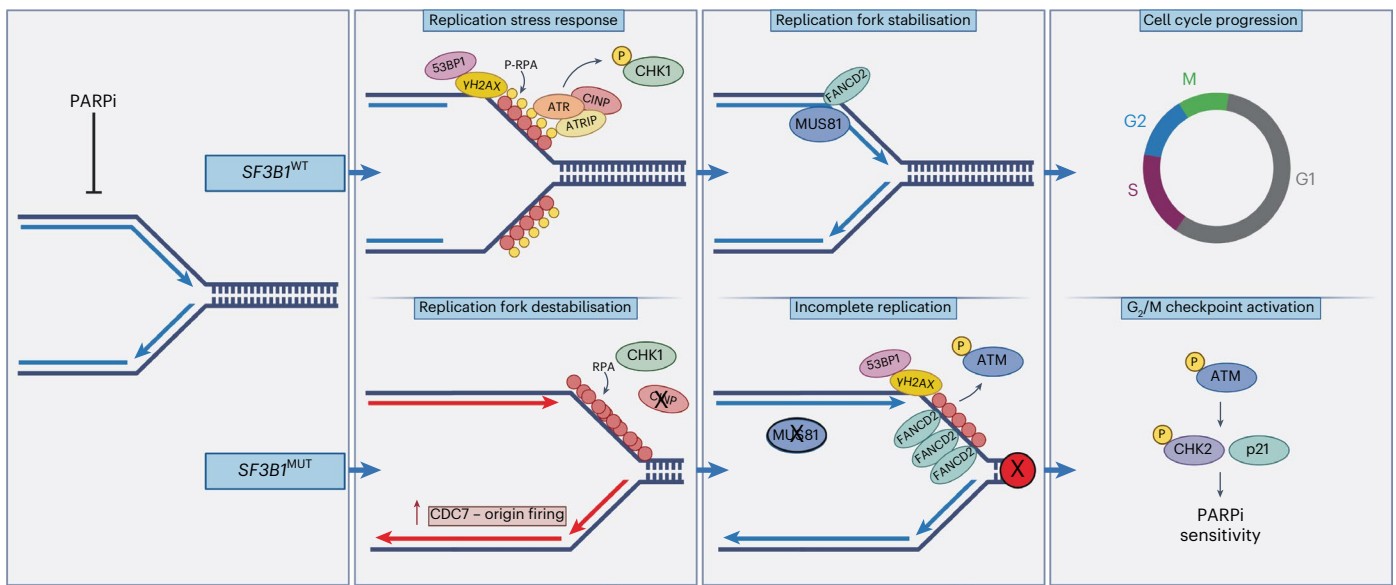

**Fig. 8 | Graphical schematic of the mechanism of PARPi sensitivity in *SF3B1*<sup>MUT</sup> cells.** When exposed to PARPi, *SF3B1*<sup>MUT</sup> cells show an impaired replication stress response (lack of pCHK1 (S317), pATR and pRPA32) due to reduced CINP protein expression. This leads to increased origin firing, unchecked fork progression and unresolved replication intermediates via the lack of MUS81-positive FANCD2 foci. This results in ATM activation and the induction of pCHK2 (T68), stalling *SF3B1*<sup>MUT</sup> cells at the G$_2$/M checkpoint.

tumor-bearing mice only, showed a significant reduction in their tumor volume, extended their survival, significantly prevented liver metastasis in 93% (14/15) of mice and, similar to the in vitro studies, induced pCHK2 (T68) (Fig. 7a–d, Extended Data Fig. 10a–d and Supplementary Fig. 7a–c). Cells from the SF3B1<sup>R625H</sup> patient-derived xenograft (PDX), PDX11310, grown subcutaneously in vivo corroborated that talazoparib significantly inhibited the growth of established tumors and extended the survival of mice, whereas cells grown from the *SF3B1*<sup>WT</sup> PDX, MP41, showed no significant difference in tumor volume with talazoparib treatment (Fig. 7e,f, Extended Data Fig. 10e–g and Supplementary Fig. 7d). We also observed that uveal melanoma patient-derived xenografts grown ex vivo as organoids (PDXOs), harboring R625H or R625C hotspot SF3B1 variants, showed sensitivity to talazoparib compared to SUM149 *BRCA1*<sup>WT</sup> revertant breast cancer cells[55] (Fig. 7g and Supplementary Fig. 7e). In addition, in vivo treatment of established NALM6<sup>H662Q</sup> *SF3B1*<sup>MUT</sup> tumors showed a significant response to talazoparib treatment compared to the vehicle; with no such antitumor effect in the NALM6<sup>K700K</sup> *SF3B1*<sup>WT</sup> tumors with talazoparib treatment (Fig. 7h,i and Extended Data Fig. 10h–j). Together, the effect of PARPi in this setting suggests that SF3B1/PARPi synthetic lethality could be further exploited and warrants investigation in future proof-of-concept clinical trials (Fig. 8).

## Discussion

Mutations in spliceosomal component genes are emerging as common characteristics of human cancers. Here we show that mutations in *SF3B1* confer selective sensitivity to clinically approved PARPi, irrespective of homologous repair functionality or *BRCA1/BRCA2* status. These effects portend to multiple molecularly diverse tumor models, supporting the hypothesis that *SF3B1* mutation status, independent of genomic background, is a determinant of sensitivity to PARPi in cancer. Given that PARPi are already approved for multiple cancer types with homologous repair defects, biomarker-driven proof-of-concept trials could be instigated to assess this hypothesis in treatment-refractory patients.

Mechanistically, these data represent a paradigm shift away from the current dogma that homologous recombination defects are the only cause of PARPi sensitivity, and implicate the largely uncharacterized protein CINP, as a major player of the replication stress response. PARP inhibition increased replication fork origin firing, resulting in accelerated fork progression in *SF3B1*<sup>MUT</sup> cells, whereas *SF3B1*<sup>WT</sup> cells under the same perturbation, induced a canonical replication stress response before reinstating an unaltered fork progression. Maya-Mendoza et al.[42] previously described a mechanism by which a PARP1-p21 axis controls fork progression and upon PARP inhibition, fork acceleration and replication stress are induced, followed by RPA and a responsive ATR (pCHK1 S317) signaling. In the context of an *SF3B1* mutation, however, we show that CINP, which has been previously linked with the cells' ability to signal DNA damage, through the phosphorylation of CHK1 at S317 (ref. 38) is downregulated in multiple *SF3B1*<sup>MUT</sup> models. This failed induction of pCHK1 (S317) coincided with a reduced replication stress response and increased origin firing upon PARPi exposure. Altered fork dynamics accumulated in unresolved replication intermediates with increased FANCD2 but lacked localized MUS81. Incomplete replication upon PARPi exposure in *SF3B1*<sup>MUT</sup> cells suggests a role for CINP in the replication stress response. Here *SF3B1*<sup>MUT</sup> cells induce ATM signaling via pCHK2 (T68) induction and likely fail to promote PLK1 (ref. 56), which ultimately stalls the cells in G$_2$/M. This PARPi-induced stalling of the cell cycle renders *SF3B1*<sup>MUT</sup> cells additionally sensitive to ATM inhibition. Targeting G$_2$/M checkpoint activation increased sensitivity to PARPi exposure and has the clinical rationale of limiting persistent cells, which have been linked to transient resistant states under PARPi treatment[57].

Of note, we did not identify any mis-splicing event in *CINP* itself, mRNA downregulation or changes in protein stability that may explain CINP downregulation, although we cannot rule out pleiotropic mis-splicing events that may act in concert to regulate CINP protein levels. Hence the exact mechanism of CINP downregulation in *SF3B1*<sup>MUT</sup> cells remains to be elucidated.

Here we note that our findings suggest a clinical utility for approved PARP-trapping agents outside the context of homologous repair-deficient cancers. Analysis of the recent PiCCLe multicenter phase 1 trial in relapsed leukemia highlighted that the patients harboring *SF3B1* mutations had the longest progression-free survival when treated with olaparib. Furthermore, we confirm CINP protein is downregulated in these *SF3B1*<sup>MUT</sup> patients[39]. Although the numbers in this study were small and at the time of writing no other trial

has reported the clinical efficacy of PARP inhibition in a homologous recombination-proficient population where *SF3B1* mutation status is known, these data suggest that PARPi treatment may have clinical benefit in this patient population. Additionally, given that a recent phase 1 clinical trial has reported no complete or partial responses in *SF3B1*[MUT] cancers treated with H3B-8800, an oral small molecule that binds SF3B1 (ref. 29), our findings are very timely, and suggest a wider group of patients with *SF3B1*[MUT] cancers, otherwise resistant to conventional treatments, may benefit from PARP-trapping drugs.

## Online content

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

[1]The Breast Cancer Now Toby Robins Research Centre, The Institute of Cancer Research, London, UK. [2]Division of Cancer Biology, The Institute of Cancer Research, London, UK. [3]Biological Services Unit, The Institute of Cancer Research, London, UK. [4]The Cancer Research UK Gene Function Laboratory, The Institute of Cancer Research, London, UK. [5]University Hospitals Birmingham NHS Foundation Trust, Birmingham, UK. [6]Institute of Cancer and Genomic Sciences, University of Birmingham, Birmingham, UK. [7]Liverpool Ocular Oncology Research Group, Department of Molecular and Clinical Cancer Medicine, University of Liverpool, Liverpool, UK. [8]Inserm U830, PSL University, Institut Curie, Paris, France. [9]Present address: Stem Cells and Cancer Division, The Walter and Eliza Hall Institute of Medical Research, Melbourne, Victoria, Australia. [10]Present address: Translational Cancer Metabolism Team, Centre for Tumour Biology, Barts Cancer Institute, Cancer Research UK Centre of Excellence, Queen Mary University of London, Charterhouse Square, London, UK. ✉e-mail: rachael.natrajan@icr.ac.uk

## Methods

All research described complies with all relevant ethical regulations. The in vivo studies carried out at The Institute of Cancer Research were performed to ARRIVE guidelines and regulations as described in the UK Animals Scientific Procedures Act 1986 and according to the UK Home Office projected licenses held by CJL and approved by the ethics board at The Institute of Cancer Research (maximum tumor size, 15 mm diameter). Additional in vivo studies were performed to local regulatory guidelines at Institut Curie (MP41 and MEL202$^{R625G\text{-}DEG}$; CEEA-IC 118, authorization APAFiS 25870-2020060410487032-v1 given by National Authority; maximal tumor volume, 2,500 mm$^3$) and Crown Biosciences (PDX11310; maximum tumor size, 2,000 mm$^3$). The maximal tumor size was not exceeded. Patients that provided samples, from which PDX were generated, were appropriately and fully consented. Additional methods are detailed in Supplementary Methods (Supplementary Information).

### Cell lines

K562 SF3B1$^{WT}$, SF3B1$^{K700K}$, SF3B1$^{K666N}$, SF3B1$^{K700E}$; NALM-6 SF3B1$^{WT}$, SF3B1$^{K700K}$ and SF3B1$^{H662Q}$, SF3B1$^{K700E}$, SF3B1$^{K666N}$ engineered isogenic cell lines were obtained from Horizon Discovery[27]. K562 and NALM-6 cells were maintained in IMDM and RPMI-1640 (Gibco), respectively, supplemented with heat-inactivated FBS (Gibco) and 1% penicillin/streptomycin. MEL202 parental SF3B1$^{R625G}$ and SF3B1$^{WT}$ revertant cell lines (MEL202$^{R625G\text{-}DEG}$), and MP46$^{WT}$, MEL270$^{WT}$ and MP41$^{WT}$ patient-derived SF3B1$^{WT}$ uveal melanoma cells[58] were cultured in RPMI-1640. MEL202$^{R625G\text{-}DEG}$ cells were engineered using the Degron-knock-in approach to harbor a degradable tag on the *SF3B1*$^{MUT}$ allele as described[25,59]. Shield-1 powder (Takara) was dissolved in 100% ethanol at 1 mmol l$^{-1}$ and stored at −20 °C. Shield-1 (Takara) was added to the fresh tissue culture media immediately before usage. All cell lines were tested regularly to confirm no mycoplasma infection using the MycoalertTM Mycoplasma Detection Kit as per the manufacturer's instructions (Lonza). Cells were authenticated by short tandem repeat typing with the Geneprint10 Kit (Promega) and were sequenced to check the retention of engineered alterations during culture. Authentication testing was last performed for all cell lines in July 2022.

### Small-molecule drug screen

The high-throughput small-molecule drug screen was performed as previously described[60], using an in-house curated 80 compound drug library present at concentrations (0.5, 1, 5, 10, 50, 100, 500 and 1,000 nM; Supplementary Table 1, Supplementary Methods). A total of 250 cells were seeded in each well of a series of 384-well plates. Twenty-four hours later, cells were exposed to small molecules and then continuously cultured for 5 d at which point cell viability was determined using Cell-Titer Glo (Promega). Survival fractions relative to DMSO controls for each drug concentration were calculated and LFC was plotted in GraphPad Prism v9.

### Splice variant analysis by qPCR

The analysis of alternatively spliced exons was performed using 384-well plates using SYBR Green (Invitrogen), (Supplementary Information, Supplementary Methods). Primers are listed in Supplementary Table 8.

### DNA-fiber analysis

For unperturbed fork dynamics, cells growing in media were incubated in medium containing 25 μM iododeoxyuridine (IdU) for 20 min, followed by 125 μM chlorodeoxyuridine (CldU) for 20 min[61]. To investigate the effect of talazoparib on DNA replication dynamics, cells growing in media were pre-incubated with 500 nM talazoparib ±20 μM CDC7i (Selleckchem, XL413) for 3 h before incubation with IdU, followed by CldU. This dose of talazoparib was chosen to ensure a robust induction of replication stress, as previously described for *BRCA*$^{WT}$ cells[42].

Fork symmetry was analyzed by calculating the ratio of the leftward and rightward tracts emanating by sister forks emerging from the same replication origin; A/B ratio > 1 indicates fork asymmetry and likely increased fork stalling. To investigate replication fork progression in conditions of exogenous induction of replication stress, cells were incubated with IdU for 30 min, followed by incubation with CldU and 100 μM hydroxyurea for 1 h. Fibers were produced from $4 \times 10^5$ cells, spread and stained as previously described with modifications; slides were blocked in 5% BSA–PBS for 30 min before primary antibody incubation with 1:20 mouse anti-BrdU (BD Biosciences, 347580) and 1:400 rat anti-BrdU (Abcam, ab6326). Before mounting of slides, slides were immersed in 70% ethanol, and then 100% ethanol. Slides were then imaged on a confocal microscope (Leica SP8) with ×63 oil objective. Analysis was performed with ImageJ software. A minimum of 300 fibers or 60 sister fork pairs were scored over at least three independent experiments. Tract lengths were measured inclusive of both IdU and CldU labeled tracts. To determine levels of origin firing, a minimum of 400 replication structures were scored across three independent experiments. The following structure classes were counted: ongoing forks (red-green tracts), origins (fired during IdU pulse green-red-green tracts or during CldU pulse green only tracts), terminations (red-green-red tracts), stalled forks (red only tracts) and interspersed forks (red-green-red-green tracts), and percentage of origins among all the structures was calculated in each of the experiments; data represent mean ± s.e.m. The raw data for each DNA-fiber measurement are provided in the Source Data and additional images are provided in Supplementary Fig. 1.

### Immunofluorescence

Before 24 h of the drug addition, adherent cell lines were seeded on glass coverslips in a multiwell plate at a density of 50,000–100,000 cells per well. Suspension cell lines were seeded in T-25 cell culture flasks at a density of $1 \times 10^6$ cells per flask and fixed in 4% paraformaldehyde in PBS for 10 min followed by three washes in PBS. Suspension cell lines were attached to glass slides using Cytospin centrifugation for 3 min at 500$g$ following fixation. The cells were then permeabilized in 0.5% Triton X-100 in PBS followed by three washes in PBS. The cells were incubated overnight at 4 °C in the primary antibody at 1/1,000 dilution in 1.5% filtered FBS in PBS. For staining of RPA and pRPA32 foci, cells were pre-extracted in ice-cold pre-extraction buffer (10 mM HEPES pH 7.5, 300 mM sucrose, 100 mM NaCl, 1.5 mM MgCl$_2$ and 0.5% Triton X-100) for 2 min before fixing. The cells were washed in PBS three times and then incubated in fluorescently labeled secondary antibodies and DAPI, diluted 1/2,000 and 1/5,000, respectively, in 1.5% filtered FBS in PBS for 60 min in the absence of light. The cells were washed twice in PBS and then mounted on glass slides with Dako Fluorescence Mounting Medium (Agilent). The slides were imaged on a Leica SP8 Confocal Microscope and quantified using CellProfiler (v3.1.9). Foci were counted using the 'speckle counting' pipeline, while phospho-histone H3, Cajal Body, p21 and nuclear area analysis was performed using the 'cell/particle counting and scoring the percentage of stained objects' pipeline. Mitotic phase analysis of the MEL202$^{R625G\text{-}DEG}$ and MEL202$^{R625G}$ cell lines was imaged using the Zeiss Axio Observer Z1 Advanced Marianas Microscope attached with a CSU-W1 SoRa and quantified by eye. The details of antibodies and buffers used can be found in Supplementary Table 9.

### Cellular viability assays

All short-term survival assays utilized 96-well cell culture plates, into which low passage, exponentially growing cells were seeded at a density of 1,000–4,000 cells per well. The drug was added 24 h post-seeding and left for 5 d of continuous exposure. Cellular viability was assessed by CellTitre-Glo luminescence assay (Promega). For clonogenic long-term assays, suspension cells were seeded in six-well plates, coated in Rat tail collagen I. NALM-6$^{WT}$ and NALM-6$^{K700K}$ (3,000 cells

per well), NALM-6[H662Q] cells (3,500 cells per well); K562[WT] (300 cells per well); K562[K700K] and K562[K700E] (650 cells per well). MEL202[R625G] and MEL202[R625G-DEG] cells were seeded in standard 6-well plates at 3,500 cells per well and SUM149 cells at 2,000 cells per well. The drug was given 24 h postseeding and to maintain a constant exposure for 14 d and fresh media with inhibitor was replaced every 72 h. For the clonogenic assay, NALM-6 and K562 cell lines were imaged without fixation and quantified on MATLAB vR20018b(9.5.0). For adherent cell lines, the colonies were solubilized with acetic acid and stained with sulphorho-damine B (Sigma-Aldrich), before measuring the optical density at 570–590 nm. Visualization of data was obtained by plotting a line of best fit to 4-parameter nonlinear regression using GraphPad Prism 9 software.

### Ex vivo talazoparib efficacy studies

The efficacy of talazoparib treatment on organoid models (ex vivo, 3D Matrigel assay) for the selected PDO models, SUM149 cell lines and the subsequent PDX11310 treatment in vivo study was carried out by Crown Bioscience San Diego (Supporting Information, Supplementary Methods).

### In vivo talazoparib efficacy studies

The NALM-6, MEL202[R625G], MEL202[R625G-DEG] and MP41 in vivo studies were performed by injecting cells subcutaneously in PBS:Matrigel (1:1; Corning Life Sciences) into 7–8 week female CB-17 (NOD. CB17-*Prkdc^scid*/J)- NALM-6 and NSG-nude mice (NOD.Cg-*Foxn1^em1Dvs Prkdc^scid* Il2rg^tm1Wjl/J)- MEL202 and MP41. To assess the tumor growth rates of NALM-6[K700K] and NALM-6[H662Q] cell lines under talazoparib treatment in CB-17 mice, treatment was given through oral gavage, with a 5 on 2 off routine at 0.33 mg kg⁻¹. A total of $2 \times 10^7$ cells were injected and when tumors averaged 100 mm³, mice were randomized and treatment commenced. For the MEL202 in vivo study, tumor growth rates and liver metastases of the MEL202 cell line with talazoparib treatment were assessed. A total of $8 \times 10^6$ cells (MEL202[R625G]) and $1 \times 10^6$ cells (MEL202[R625G-DEG]) were injected subcutaneously into NSG-Nude mice and when tumors averaged 100 mm³, mice were randomized and underwent treatment. Treatment was given through oral gavage, daily, at a concentration of 0.33 mg kg⁻¹. For both studies, the Solutol-based vehicle was 10% DMAc, 6% Solutol and 84% PBS, DMSO controls were also diluted in the vehicle, tumors were measured 2/3 times a week with calipers and mice were weighed twice a week. Studies were terminated when control arm measurements neared but were less than 15 mm in diameter, in any direction, and statistical analysis was performed using Prism. The PDX model MP41 was treated with the PARPi talazoparib in vivo at the Institut Curie. Tumor fragments of 15 mm³ were trans-planted into NSG-nude mice and animals were randomized when the tumor volume reached 100 mm³ and treated with vehicle (10% DMAc, 6% Solutol and 84% PBS; Group 1) or talazoparib (0.33 mg kg⁻¹; Group 2) and approved by local ethics. Groups 1 and 2 were killed on day 28. The PDX model PDX11310 was treated with the PARPi talazoparib in vivo in 7- to 8-week-old female NOD-SCID (NOD.Cg-*Prkdc^scid*/J) mice by Crown Bioscience. Animals were randomized when the tumor volume reached 150–250 mm³ and treated with vehicle (Group 1; 10% DMAc, 6% Solutol and 84% PBS) or talazoparib (0.33 mg kg⁻¹; Group 2) and approved by local ethics. Groups 1 and 2 were euthanized on day 31. End-of-study tumors were taken for fixed and snap-frozen samples. Tumor cDNA and gDNA from each animal were taken and sequenced to check for the retention of the SF3B1[R625H] variant, originally denoted in this PDX model. Tumors were formalin-fixed and paraffin-embedded (FFPE), and sections were stained with hematoxylin and eosin (H&E), or incubated with antibodies against Ki-67 and cleaved caspase-3.

### Cell cycle analysis

Cell cycle analysis was undertaken using propidium iodide (Abcam, ab14083) and analyzed on BD LSRII cell analyzer. Trypsinized cells were washed twice in PBS before fixation through the dropwise addi-tion of 70% ethanol and allowed to fix for 30 min at 4 °C. Cell pellets were washed twice with PBS at 850*g*, treated with 50 µl of 100 ug ml⁻¹ RNase and resuspended in 200 µl of 50 µg ml⁻¹ propidium iodide. Forward and side scatters were set to identify single cells and dou-blets were excluded. Gates were then automatically set and percent-ages were derived by use of FlowJO v10.8 (BD Biosciences) analysis software.

Cell cycle reporter cell lines, MEL202[R625G] and MEL202[R625G-DEG] were generated with the Incucyte Cell Cycle Green/Red Lentivirus Reagent (EF1α-Puro; Satorius 4779), at an MOI of 0.03 transduction units (TU) per cell, and cultured in 2 µg ml⁻¹ puromycin (Gibco) for 21 days to isolate and amplify stable clones. Stably transfected cells were plated in 12-well plates at a density of 100,000 cells per well. Twenty-four hours post seeding, cells were treated with talazoparib and imaged at 1 h intervals on the Incucyte S3 Live-Cell Analysis System (Sartorius). Red, green and yellow fluorescent cells were quantified using the built-in analysis to calculate the cell cycle profile.

### Paired-end RNA sequencing

RNA sequencing of K562 SF3B1[WT] and SF3B1[K700E] cell lines was per-formed using 100 ng of ribosomal-depleted RNA from cell lines grown in triplicate from independent passages and treated with 100 nM tala-zoparib for 48 h. RNA libraries were prepared using the NEBNext Ultra Directional RNA Library Preparation Kit according to the manufac-turer's instructions, with 200 bp fragments size selection and eight cycles of PCR amplification, and were sequenced on a single lane of a HiSeq 2500 using SBS v3 chemistry (Illumina; 2 × 100 bp cycles), result-ing in >40 million paired end-reads. RNA sequencing FASTQ files were aligned to the human genome (hg38) using STAR v2.5.1b[62] with the addi-tional custom parameters '--twopassMode Basic --outSAMstrandField intronMotif --outSAMattributes NH HI AS nM NM XS' with transcript annotations obtained from GENCODE v22. Differential gene expres-sion analysis was performed using a negative binomial generalized log-linear model (glmQLFit and glmQLFTest) implemented in edgeR v3.34.0 (ref. [63]). Normalization factors to correct for variable sequenc-ing depth and composition bias were calculated using the trimmed mean of M-values (TMM) method[64] (calcNormFactors). GSEA was per-formed with FGSEA[65] v1.4.1 using the c2.cp.reactome gene sets obtained from the Broad Institute with the minimum pathway size set to 10. Genes were ranked according to $-\log_{10}$(raw $P$ value) multiplied by the sign of the $\log_2$ fold change. Quantification of PSI (Ψ) (percentage spliced in) values for the alternative splicing event types (alternative 5′, alternative 3′, exon skip, multiple exon skip and intron retention) was performed with spladder (development version dated 3 July 2018)[66] under default settings (confidence level = 3). Additional filtering required at least five supporting and excluding junction reads in at least 25% of samples to remove under-represented events. rMATS v4.1.2 (ref. [67]) was run under default parameters. Detection of differential alternative splicing events from both spladder and rMATS between K562 SF3B1[WT] and SF3B1[K700E] cells was assessed by performing a differential PSI (Ψ) analysis using the limma v3.48.3 package[68], Benjamini–Hochberg adjusted $P < 0.1$. Sequence motif logos illustrating 30 bp upstream and 3 bp downstream of significant alternative 3′ acceptor splice sites were generated using ggseqlogo v0.1 (ref. [69]). For visualization purposes, the most sig-nificant events (Benjamini–Hochberg adjusted $P < 0.01$ and $|\Delta\Psi| > 5\%$) were selected. Raw RNA-sequencing data are publicly available through SRA accession number PRJNA968072.

### Total mass spectrometry and proteomic profiling

Cell lines were treated with DMSO or talazoparib at 50 nM for 48 h and cell pellets were lysed in 5% SDS per 100 mM TEAB buffer with probe sonication and heating at 95 °C. Further, 57 µg of protein was reduced with TCEP and alkylated by iodoacetamide followed by TCA (trichlo-roacetic acid) precipitation and digested overnight in Trypsin at 37 °C

(MS grade, Thermo Fisher) was added at 1:25 (trypsin:proteins). Peptides were TMT labeled as instructed by the manufacturer, then mixed, SpeedVac dried and fractionated on a BEH XBridge C18 column (2.1 mm i.d. × 150 mm) with a 35 min gradient from 5–35% $CH_3CN/NH_4OH$ at pH 10. A total of 36 fractions were collected and SpeedVac dried, then resuspended in 0.5%FA/$H_2O$, and 50% was injected for LC–MS/MS analysis on an Orbitrap Fusion Lumos coupled with an Ultimate 3000 RSLCnano System.

Samples were loaded on a nanotrap (100 µm id × 2 cm; PepMap C18, 5 µ) at 10 µl min⁻¹ with 0.1% formic acid and then separated on an analytical column (75 µm id × 50 cm; PepMap C18, 2 µ) over at 300 nl min⁻¹ at a 90 min gradient of 4–30.4% $CH_3CN$/0.1% formic acid per 120 min cycle time per fraction.

Raw files were processed with Proteome Discoverer 2.3 (Thermo Fisher) and searched using both SequestHT and Mascot (v2.3 MatrixScience) against UniProt Human Reference Proteome database (January 2018) concatenated with the cRAP contaminate sequences (precursor mass tolerance, $t = 30$ ppm; fragment ion mass tolerance, 0.5 Da). Spectra were searched for fully tryptic peptides with a maximum of two miss-cleavages. Target/decoy peptides were processed with Percolator and the consensus search result was filtered to a protein false discovery rate adjusted (FDR) of 0.01 (strict) and 0.05 (relaxed). The TMT10plex reporter ion quantifier used 20 ppm integration tolerance on the most confident centroid peak at the MS3 level. Only unique peptides with average reported S/N > 3 were used for quantification. Only master proteins for each peptide group were reported.

### RNA polymerase II ChIP–sequencing

K562 isogenic cell lines were submitted to Active Motif for ChIP–seq for total RNA Pol II using 30 µg input chromatin (RNA Pol II antibody Active Motif 39097). The 75-nt sequence reads generated by Illumina sequencing (using NextSeq 500) were mapped to the hg38 reference genome using BWA algorithm vv0.7.12 with default settings. Only reads passing Illumina's purity filter, aligned with no more than two mismatches and mapped uniquely to the genome were used. Peaks were called using SICER v1.1 (ref. [70]) FDR of $1 × 10^{-10}$ with a gap parameter of 600 bp. Peak filtering was performed by removing false ChIP–seq peaks as defined within the ENCODE blacklist[71]. Merged regions were computed (genomic regions containing 1 or multiple overlapping intervals) to allow comparisons between samples. Peak ratios of the intersect of LFC >|1| K700E versus wild-type and LFC >|1| K700E versus K700K were considered differential.

### Statistical analyses

Statistical analysis was carried out using R 3.5.0 (www.r-project.org) and GraphPad Prism 9. Comparisons between groups of continuous variables were made using an unpaired two-tailed Student's $t$-test, Mann–Whitney $U$ test, Welchs' $t$-test or ANOVA. Univariate differences in survival were analyzed by the Kaplan–Meier method and significance was determined by the log-rank test. All tests were two-sided and a $P$ value of less than 0.05 was considered significant. FDR $P$ values for multiple testing were used for RNA-sequencing and proteomic analyses, with an FDR value of <0.1 considered significant (unless otherwise indicated). Pathway enrichment of the proteomic data was performed with GSEA v1.18.0 (ref. [72]), on a preranked list of genes sorted by their PARPi versus DMSO log₂ fold change. The number of permutations was set to 10,000 and the adjusted (FDR) $P$ value cut-off was set to 0.05. The numbers of independent biological replicates are included in each figure legend as are details of the numbers of events counted. No animals were excluded from the in vivo analyses. Tumor volume data points from in vivo studies were excluded on the rare occasion the measurements were inaccurate (that is, the mice had skin thickening over the inoculation site or was not measurable on that day) as detailed in the Source Data. No data points were excluded from other experiments.

### Reporting summary

Further information on research design is available in the Nature Portfolio Reporting Summary linked to this article.

### Data availability

The data that support the findings of this study are available in the Supporting Information. The RNA sequencing data have been deposited in NCBI Sequence Read Archive (SRA) under accession number PRJNA968072; ChIP–seq data PRJNA968071 and the mass spectrometry proteomics data have been deposited to the ProteomeXchange Consortium via the PRIDE partner repository with the dataset identifier PXD019046. *SF3B1* mutations were collated from cBioPortal https://www.cbioportal.org/ querying MSK-IMPACT PanCancer Clinical Sequencing cohort and TCGA Pan-Cancer Atlas studies. Database access July 2020. UniProt Human Reference Proteome database (January 2018) was used as a reference for the mass spectrometry data. Source data are provided with this paper.

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

## Acknowledgements

This study was supported by Breast Cancer Now as part of program funding to the Breast Cancer Now Toby Robins Research Centre (to R.N. and C.J.L.); The Institute of Cancer Research (to R.N.); and MRC Confidence in Concept award (MC_PC_16047 to R.N.); CRUK programmatic funding (to C.J.L.), CRUK program grant (C20807/A28640 to T.S. and C.O.); ICR intramural grant and Cancer Research UK program (A24881 to W.N.); Wellcome Trust 20839 (to J.C.). A.H. is supported by Site de Recherche Intégrée sur le Cancer (SiRIC2) de l'Institut Curie. The authors thank S. Dayot for generating the MEL202$^{R625G\text{-}DEG}$ cell line. We thank V. Pena, I. Bajrami, J. Frankum, D. Krastev, F.-F. Song and D. Zatreanu for helpful discussions. We thank D. Didier and F. Nemati for help with the in vivo efficacy studies of UM models. The results shown here are in part based on data generated by the TCGA Research Network (http://cancergenome.nih.gov/). We acknowledge NHS funding to the ICR/Royal Marsden Hospital Biomedical Research Centre. Figs. 1c, 8 and Extended Data Fig. 1i were created with BioRender.com.

## Author contributions

P.B., H.S., P.T.W., L.C., G.M., J.N., N.R., M.B.J., S.H., H.E.B., J.W., L.Y., I.M., A.R., B.P., M.A., P.G., H.P., A.G., S.N., F.N., N.G., I.R., G.P., C.O., T.S., S.B., H.K., S.E.C., R.B., S.A., A.H., M.H.S., S.P., J.C., S.H., W.N., C.J.L. and R.N. generated and analyzed data and/or developed methodology. P.B., H.S. and R.N. drafted the manuscript. All authors edited and approved the final submitted manuscript. P.B. and R.N. conceptualized the study. S.H., J.C., W.N., C.J.L. and R.N. helped in supervision and funding acquisition of the study.

## Competing interests

W.N. is a named inventor on a patent describing the use of EXD2 inhibitors and stands to gain from their development as part of the ICR 'Rewards to Inventors' scheme, and was a consultant for MNM Bioscience. C.J.L. makes the following disclosures: receives and/or has received research funding from AstraZeneca, Merck KGaA, and Artios; received consultancy from SAB membership or honoraria payments from Syncona, Sun Pharma, Gerson Lehrman Group, Merck KGaA, Vertex, AstraZeneca, Tango, 3rd Rock, Ono Pharma, Artios, Abingworth, Tesselate and Dark Blue Therapeutics; has stock in Tango, Ovibio, Enedra Tx., Hysplex and Tesselate. C.J.L. is also a named inventor on patents describing the use of DNA repair inhibitors and stands to gain from their development and use as part of the ICR 'Rewards to Inventors' scheme and also reports benefits from this scheme associated with patents for PARPi paid into CJL's personal account and research accounts at the Institute of Cancer Research. R.N. receives and/or has received academic research funding from Pfizer in the form of the Breast Cancer Now Catalyst academic grant scheme. AstraZeneca partially supported the PiCCLe clinical trial (supplied by Olaparib; this study is published). The remaining authors declare no conflicts of interest.

## Additional information

**Extended data** is available for this paper at https://doi.org/10.1038/s41588-023-01460-5.

**Correspondence and requests for materials** should be addressed to Rachael Natrajan.

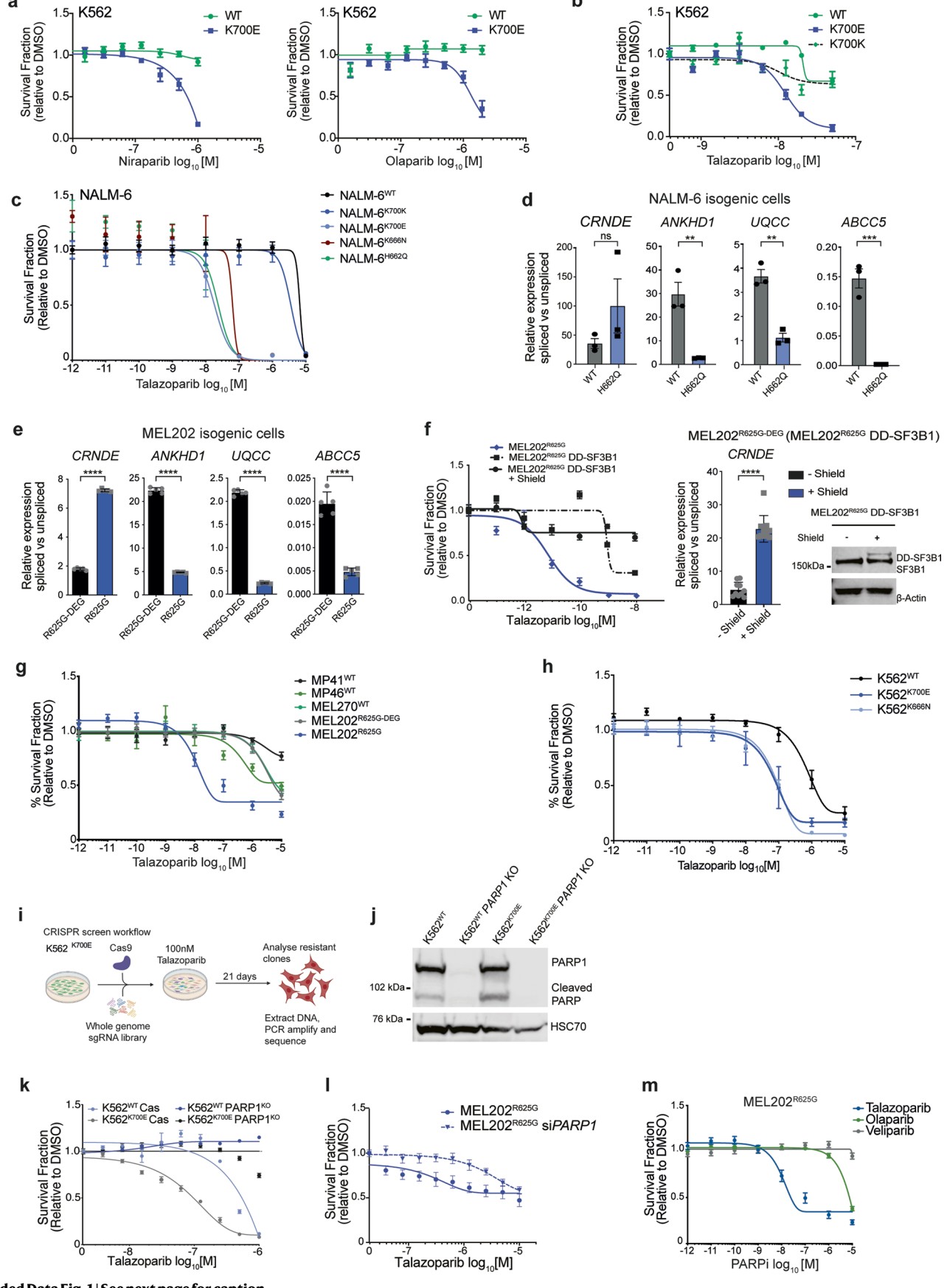

**Extended Data Fig. 1 | See next page for caption.**

**Extended Data Fig. 1 | *SF3B1* hotspot mutations induce mis-splicing and PARPi sensitivity. a**, **b**, 14 day clonogenic dose–response curves of K562 SF3B1$^{WT}$, SF3B1$^{K700K}$ (silent mutation) and SF3B1$^{K700E}$ isogenic cells following exposure with distinct PARP inhibitors. Data are mean ± s.e.m, ($n$ = 3 independent biological replicates). **c**, 14 day (3D viability) talazoparib dose–response curves of NALM6$^{WT}$, NALM6$^{K700K}$ and SF3B1$^{MUT}$ NALM6$^{K700E}$, NALM6$^{K666N}$ and NALM6$^{H662Q}$ cell lines grown as spheroids. Data are mean ± s.e.m, ($n$ = 3 independent biological replicates). **d**, Representative qRT-PCR of differentially spliced exons of indicator genes in the NALM-6$^{WT}$ and NALM-6$^{H662Q}$ isogenic lines. Data are mean of $n$ = 3 biological replicates, ± s.d. (unpaired two-tailed t-test (NS $P$ = 0.2426, **$P$ = 0.0058, **$P$ = 0.0015, ***$P$ = 0.0009)). **e**, Representative qRT-PCR of differentially spliced exons of indicator genes in the MEL202$^{R625G-DEG}$ and MEL202$^{R625G}$ cells. Data are mean of $n$ = 5 biological replicates, ± s.d. (unpaired two-tailed t-test (****$P$ < 0.0001)). **f**, 14 day clonogenic dose–response for the isogenic MEL202$^{R625G-DEG}$ and MEL202$^{R625G}$ cells exposed to talazoparib and revertant MEL202$^{R625G-DEG}$ cells labeled with a degron tag (MEL202$^{R625G}$ DD-SF3B1) +/- Shield-1 compound to stabilize expression of the mutant allele. Data are

normalized to DMSO control and presented as mean ± s.e.m. ($n$ = 3 biological replicates). qRT-PCR of differentially spliced exon of *CRNDE* in the MEL202$^{R625G-DEG}$ +/- shield compound ($n$ = 3 biological replicates, ****$P$ < 0.0001, unpaired two-tail t-test). Western blot analysis of MEL202$^{R625G-DEG}$ (MEL202$^{R625G}$ DD-SF3B1) showing protectable mutant allele upon shield compound treatment. **g**, 5-day viability dose–response curves of wild-type uveal melanoma cell lines MP41, MP46, MEL270 and MEL202. Data are mean ± s.e.m ($n$ = 3 biological replicates). **h**, 14 day (3D viability) dose–response curves of K562$^{WT}$, K562$^{K700E}$ and K562$^{K666N}$ spheroids exposed to talazoparib. Data are mean ± s.d. ($n$ = 3 independent biological replicates). **i**, Schematic of CRISPR screen workflow. **j**, Western blot of PARP1, cleaved PARP1 and HSC70 in K562$^{WT}$ and K562$^{K700E}$ cells with Cas or PARP1 KO. **k-l**, Talazoparib dose-response curves showing the survival fraction of K562 isogenic cells +/- PARP1 CRISPR knockout (KO) (k), MEL202$^{R625G}$ cells +/- *PARPi* siRNA (l). Data are mean of $n$ = 3 biological replicates, ± s.e.m. **m**, 5 day dose–response curve of MEL202$^{R625G}$ cells exposed to talazoparib, olaparib and veliparib. Data are mean ± s.e.m. ($n$ = 3 biological replicates).

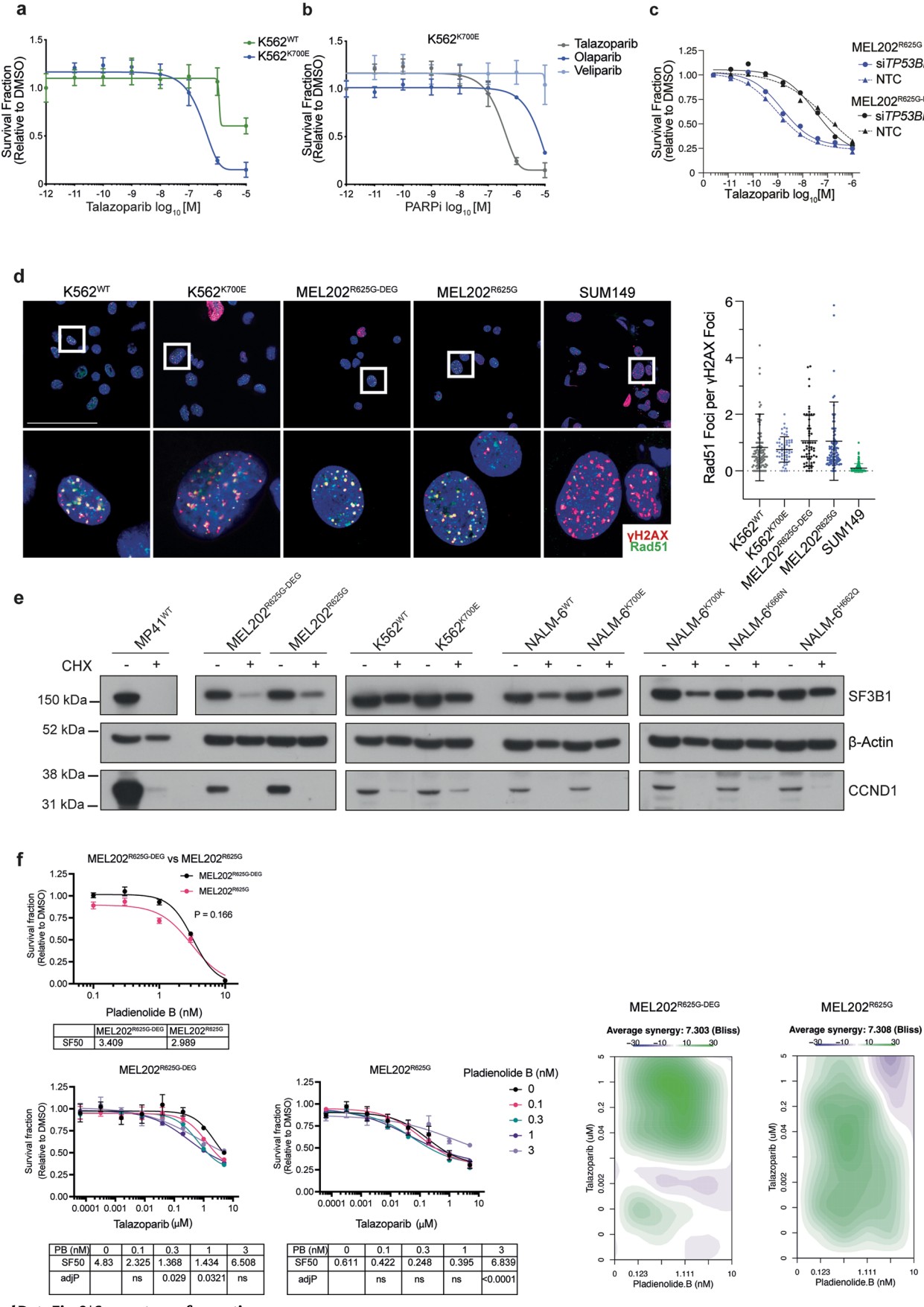

**Extended Data Fig. 2 | See next page for caption.**

**Extended Data Fig. 2 | *SF3B1* hotspot mutations induce PARPi sensitivity.**
**a**, 5 day dose–response curve of K562$^{WT}$ and K562$^{K700E}$ cells treated with
talazoparib ($n$ = 3 biological replicates). Data are presented as mean ± s.e.m.
**b**, 5 day dose–response curve of K562$^{R625G}$ cells exposed to talazoparib, olaparib
and veliparib indicating sensitivity to more potent PARP trapping agents.
($n$ = 3 biological replicates). Data are presented as mean ± s.e.m. **c**, 5 day dose–
response curve of MEL202$^{R625G}$ cells exposed to talazoparib MEL202 isogenic
cells +/- *TP53BP1* siRNA gene silencing. Data are presented as mean ± s.e.m.
($n$ = 3 technical replicates). **d**, Representative immunofluorescence images and
corresponding scatter plot graph showing the number of RAD51 foci per γH2AX
foci in K562 and MEL202 isogenic cell lines and SUM149 *BRCA1$^{MUT}$* cells after
10 Gy irradiation. Data are presented as mean ± s.e.m. ($n$ = 1 biological replicate).

**e**, Western blot of SF3B1 protein expression in UM MP41$^{WT}$ cells, MEL202, K562
and NALM-6 isogenic cell lines exposed to 50 mg/ml cycloheximide (CHX) for
48 hours. CCND1 is used as a control for protein degradation, due to a relative
short half-life. **f**, Dose–response curves of MEL202$^{R625G-DEG}$ and MEL202$^{R625G}$ cells
exposed to Pladienolide B as single agent (a) or in combination with talazoparib
(b). SF50 values of combinations at different Pladienolide B concentrations are
shown. Data are presented as mean, ± s.d. of $n$ = 3 technical replicates.
(c) Heatmaps showing BLISS synergy scores based on the survival fraction,
relative to DMSO, of MEL202 isogenic cells after 5 days of exposure to
talazoparib in combination with Pladienolide B. P values from one way ANOVA
with Tukey's multiple comparison test.

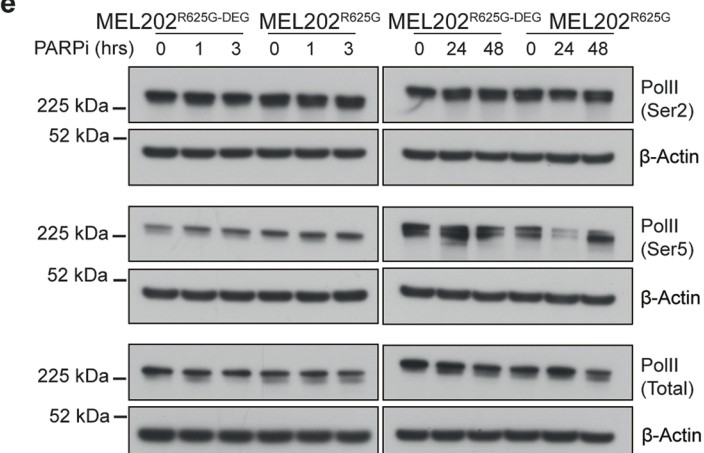

**Extended Data Fig. 3 | *SF3B1* mutant cells show transcriptional dysregulation following PARPi exposure. a**, Frequency plot of proportion of PSI events of aberrant splicing calculated from total RNA-sequencing (*n* = 3 biological replicates) of K562[WT] versus K562[K700E], with and without PARPi (a, Spladder and b, rMATS). Multiple skipped exons (MES), retained intron (IR), skipped exon (ES), alternative 5' splice site (A5) and alternative 3' splice site (A3) events with an FDR < 0.1. **b**, Splice site motif analysis of aberrant A3 events depicting canonical and alternative branch point usage in K562[K700E] versus K562[WT] cells +/- talazoparib detected from total RNA sequencing. AG represents the 3'ss and the upstream adenines (A) represent the branch points. Related to Fig. 2a. **c**, Heatmap depicting the distribution of the overall binding of RNA Pol II in K562[WT], K562[K700K] control and K562[K700E] cells. **d**, Frequency plot of RNA Pol II binding at transcription start sites in the K562[WT], K562[K700K] control and K562[K700E] cells. (*n* = 1 biological replicate). **e**, Western blot of Ser5 (initiation), Ser2 (elongation) and total RNA Pol II in MEL202[R625G-DEG] and MEL202[R625G] cells exposed to short term (0, 1, 3 hours) and long term (0, 24 and 48 hours) talazoparib alongside β-Actin loading control (*n* = 1 biological replicate).

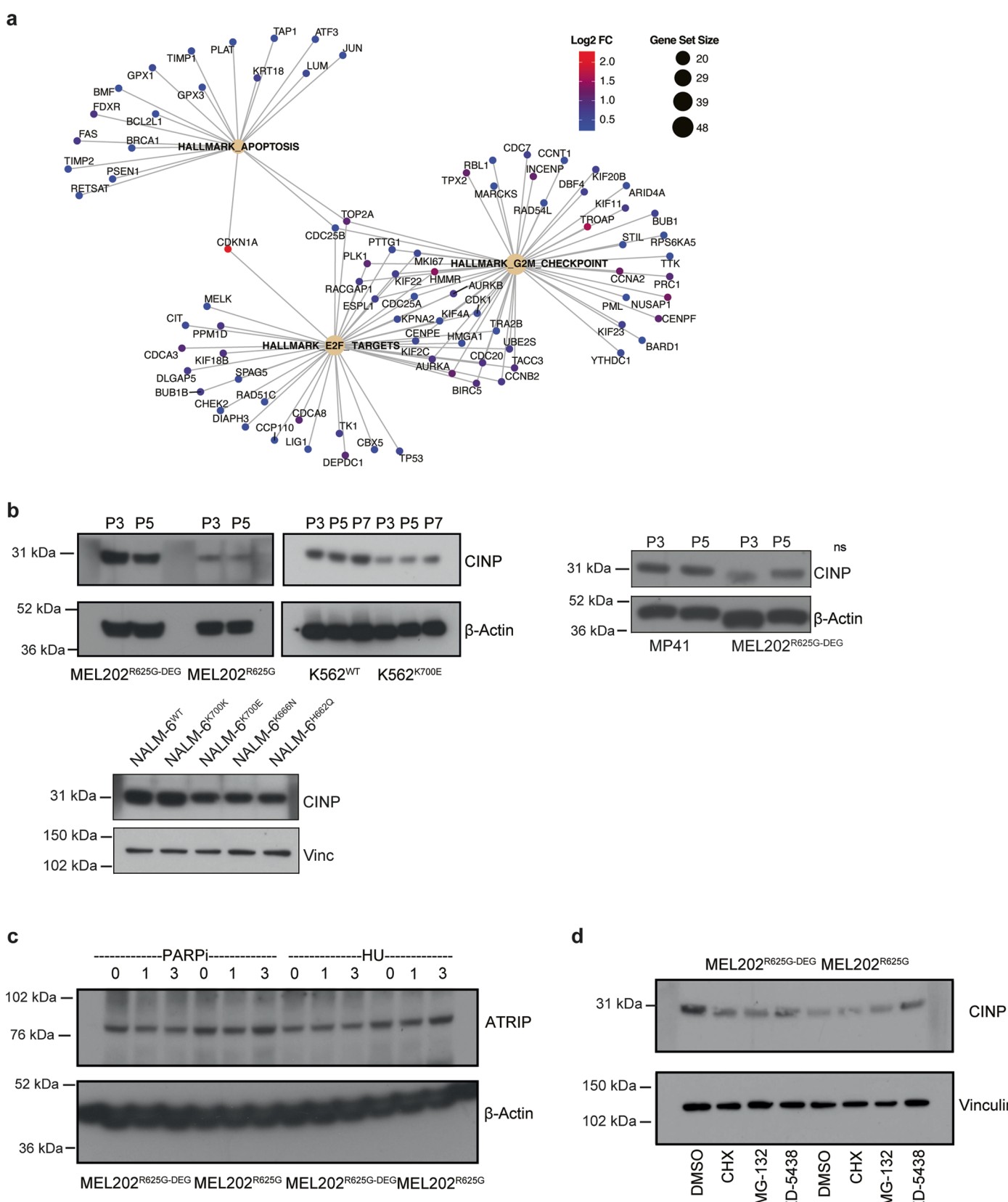

**Extended Data Fig. 4 | See next page for caption.**

**Extended Data Fig. 4 | G₂/M checkpoint protein expression in *SF3B1* mutant cells under PARPi. a**, Schematic showing the log₂FC of protein expression in the gene sets 'HALLMARK_APOPTOSIS', 'HALLMARK_E2F_TARGETS', and 'HALLMARK_G2M_CHECKPOINT', and the overlapping genes in these gene sets. Data taken from the total-MS (mass spectrometry) in Fig. 2d. **b**, Western blot of CINP and β-Actin loading control in MEL202, K562 and NALM-6 *SF3B1*^WT and *SF3B1*^MUT isogenic cell line pairs under different cell passages ('P'). **c**, Western

blot of total ATRIP and β-Actin loading control in MEL202, *SF3B1*^WT and *SF3B1*^MUT isogenic cell lines +/- PARPi talazoparib or hydroxyurea (HU) for indicated times (hours) (*n* = 1 biological replicate). **d**, Western blot of CINP expression in MEL202^R625G-DEG and MEL202^R625G cells and vinculin loading control after 6 hours exposure to DMSO, cycloheximide (CHX, 10 μM), MG-132 (20 μM) and AZD-5438 (5 μM) (CDK2i) (*n* = 1 biological replicates).

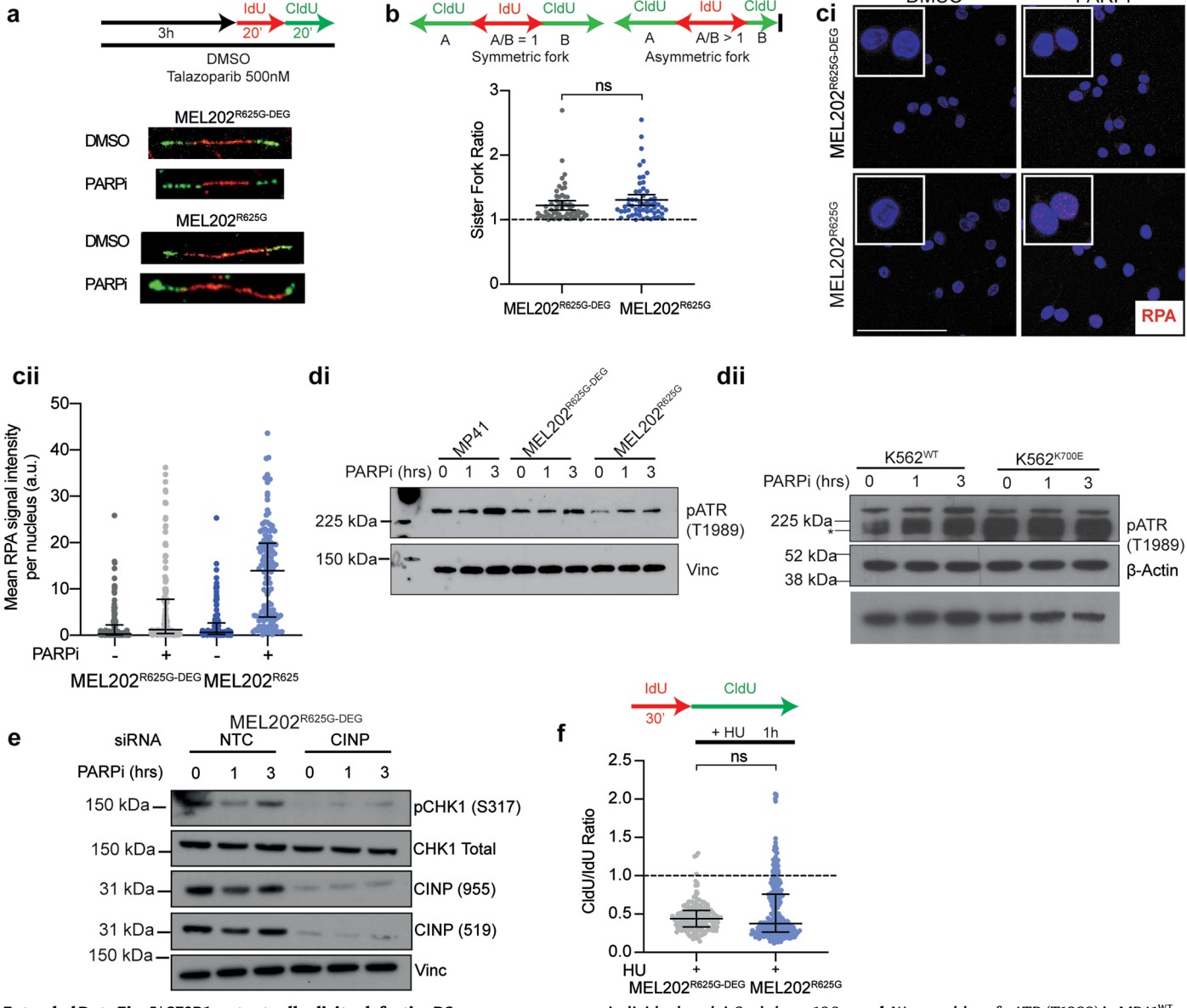

**Extended Data Fig. 5 | *SF3B1* mutant cells elicit a defective RS response under PARPi. a**, Experimental set up of fiber assay and representative immunofluorescence images of IdU and CldU labeled DNA fibers after 3 hours 500 nM talazoparib or DMSO exposure. **b**, Schematic of analysis and scatterplot of quantification of sister fork ratio taken from DNA fiber analysis of MEL202 isogenic cells exposed to DMSO. Data are mean of $n = 3$ biological replicates, error bars show ± s.e.m. (unpaired two-tailed t-test (NS $P = 0.1337$)). **c**, Representative immunofluorescence images (ci) and scatterplot (cii) of RPA foci in MEL202 isogenic cells following 3 hours of 500 nM talazoparib or DMSO exposure. Data are from $n = 2$ biological replicates, error bars show ± s.d. of foci in individual nuclei. Scale bar = 100 μm. **d**, Western blot of pATR (T1989) in MP41$^{WT}$ and MEL202 isogenic cells (di) and K562 isogenic cells (dii) at 0, 1, or 3 hours of 500 nM talazoparib exposure. **e**, Western blot of pCHK1 (S317), total CHK1, and CINP expression using two different CINP antibodies in MEL202$^{R625G-DEG}$ cells after non-targeting control (NTC) or *CINP* siRNA gene mediated silencing, at 0, 1, or 3 hours of 500 nM talazoparib exposure ($n = 1$ biological replicate). **f**, Scatterplot of CldU/IdU ratio taken from DNA fiber analysis of MEL202 isogenic cells exposed to 100 μM hydroxyurea (HU). Data are mean of $n = 3$ biological replicates, error bars show ± s.e.m. (unpaired two-tailed t-test (NS $P = 0.458$).

**a**

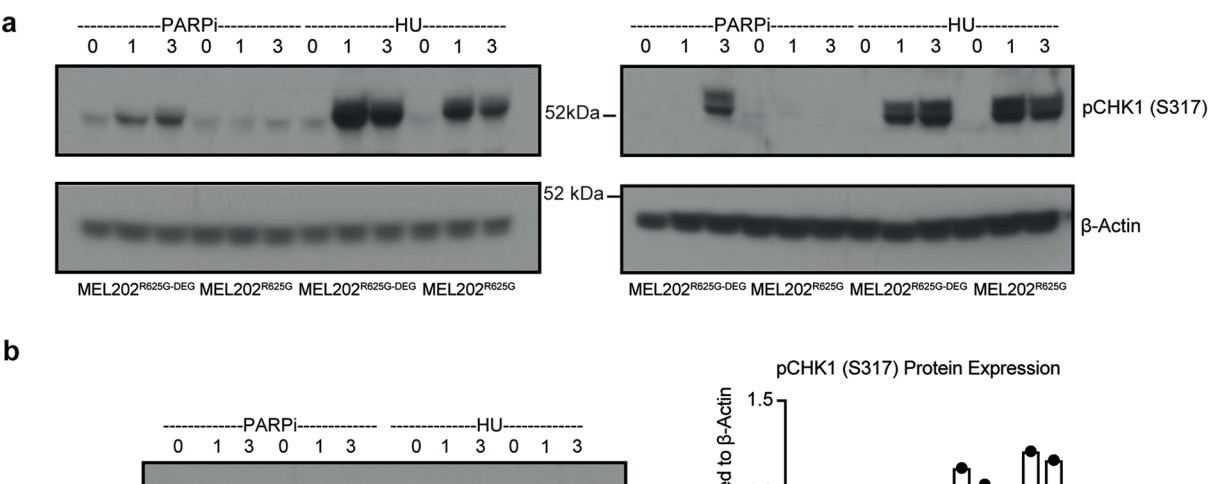

**b**

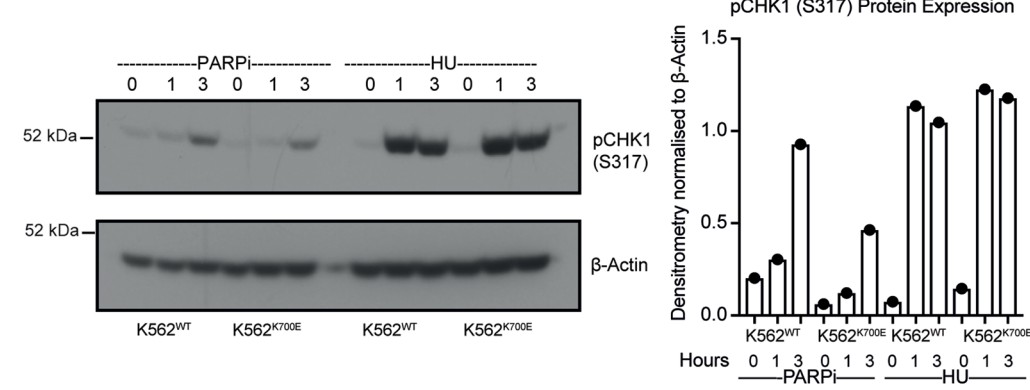

**c**

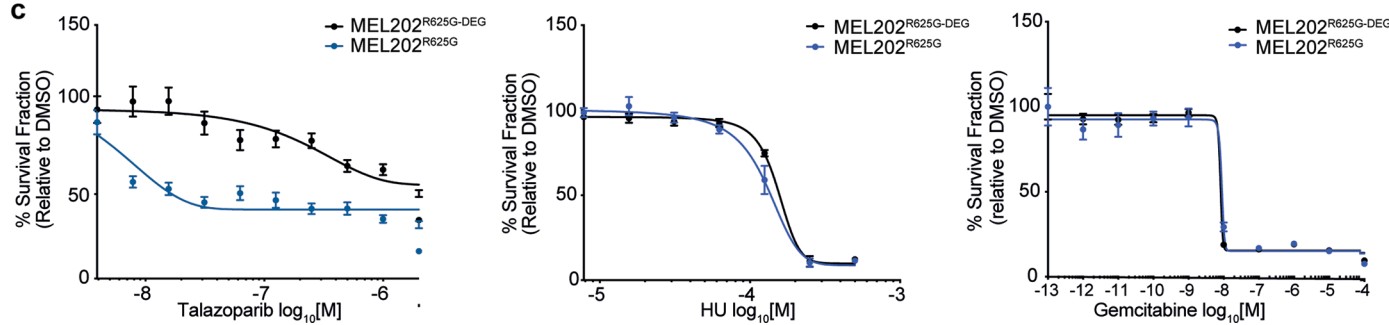

**Extended Data Fig. 6 | *SF3B1* mutant cells elicit a replication stress response upon hydroxyurea exposure. a**, **b**, Western blots of pCHK1 (S317) expression in MEL202 isogenic (a) and K562 isogenic cells (b) after 0, 1, and 3 hours of 500 nM talazoparib or 100 µM hydroxyurea (HU) exposure, and column bar graph showing relative pCHK1 (S317) expression relative to β-Actin loading control.

Images are representative of $n = 2$ biological replicates. **c**, Talazoparib, HU, and gemcitabine dose-response curves showing the survival fraction, relative to DMSO, of MEL202 isogenic cells. Data are mean of at least $n = 2$ biological replicates, error bars show ± s.d.

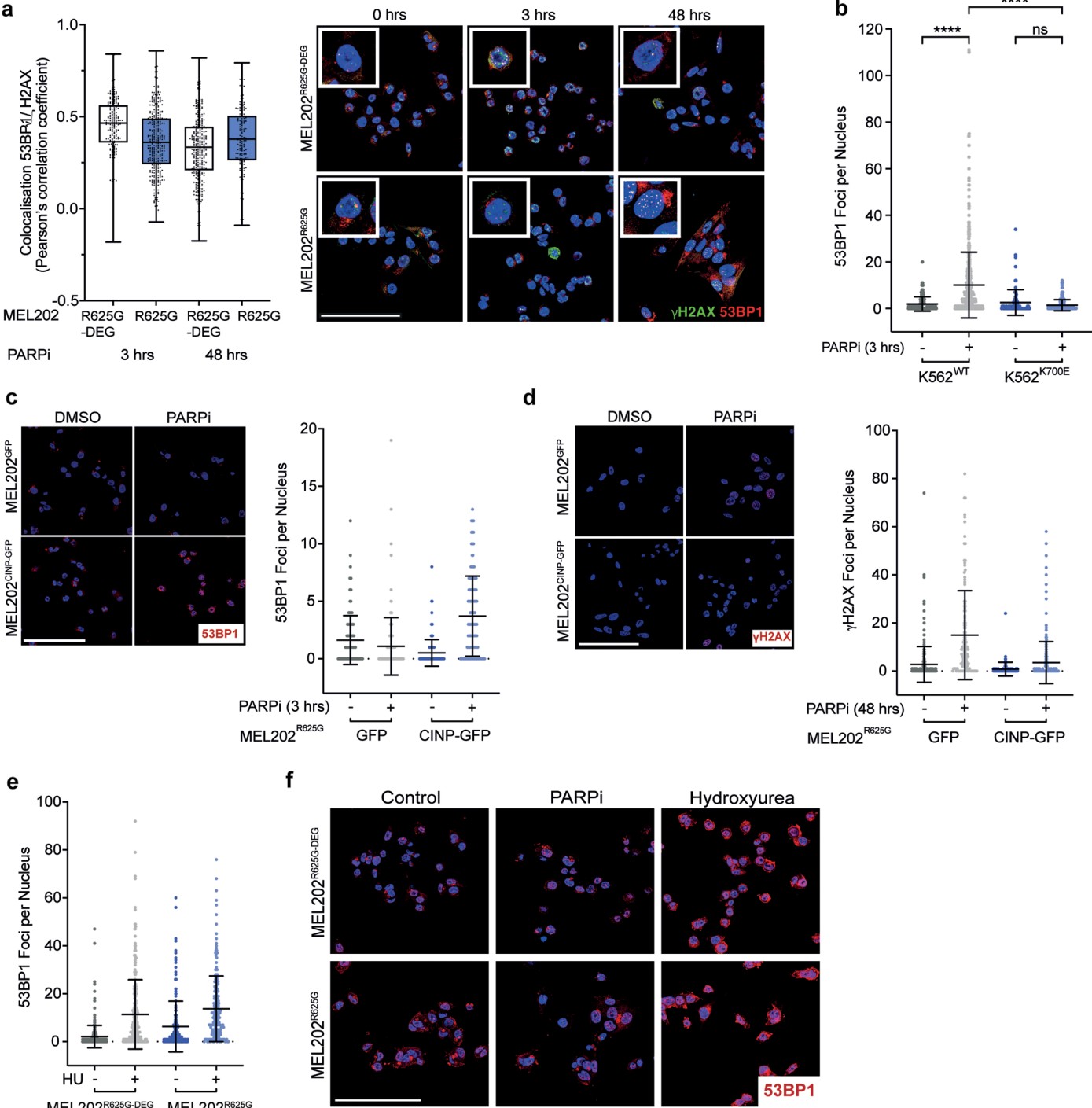

**Extended Data Fig. 7 | *SF3B1* mutant cells have a defective replication stress regulatory response upon PARPi exposure. a**, Box and whiskers plot and representative images showing the colocalization of 53BP1 and γH2AX in MEL202 isogenic cells after 3 hours of 500 nM, or 48 hours of 50 nM, talazoparib exposure. Colocalization based on the Pearson's correlation coefficient of the 53BP1 and γH2AX fluorescence intensity per nuclei (*n* > 220 cells from *n* = 1 biological replicate) Scale bar = 100 μm, error bars show ± s.d. **b**, Scatterplot showing the number of 53BP1 foci per nucleus in K562 isogenic cells after 3 hours of 500 nM talazoparib or DMSO exposure. Data are mean of *n* = 3 biological replicates, error bars show ± s.d. (unpaired two-tailed t-test (****P < 0.0001,

NS *P* = 0.238)). **c-d**, Representative immunofluorescence images and scatter plot quantification of 53BP1 (c) and γH2AX (d) foci in MEL202R625G cells expressing control-GFP or CINP-GFP, treated with 3 hours of 500 nM (c) *(n = >205 cells from n = 2 independent biological replicates)* or 48 hours of 50 nM talazoparib (d) *(n = >126 cells from n = 1 biological replicate)*. Error bars show ± s.d. **e-f**, Scatter plot quantification and representative immunofluorescence images of 53BP1 foci in MEL202 isogenic cells after 100 μM HU (n > 214 cells from *n* = 1 biological replicate), or DMSO exposure. Error bars show ± s.e.m, 500 nM talazoparib (*n* > 215 cells from *n* = 3 independent biological replicates).

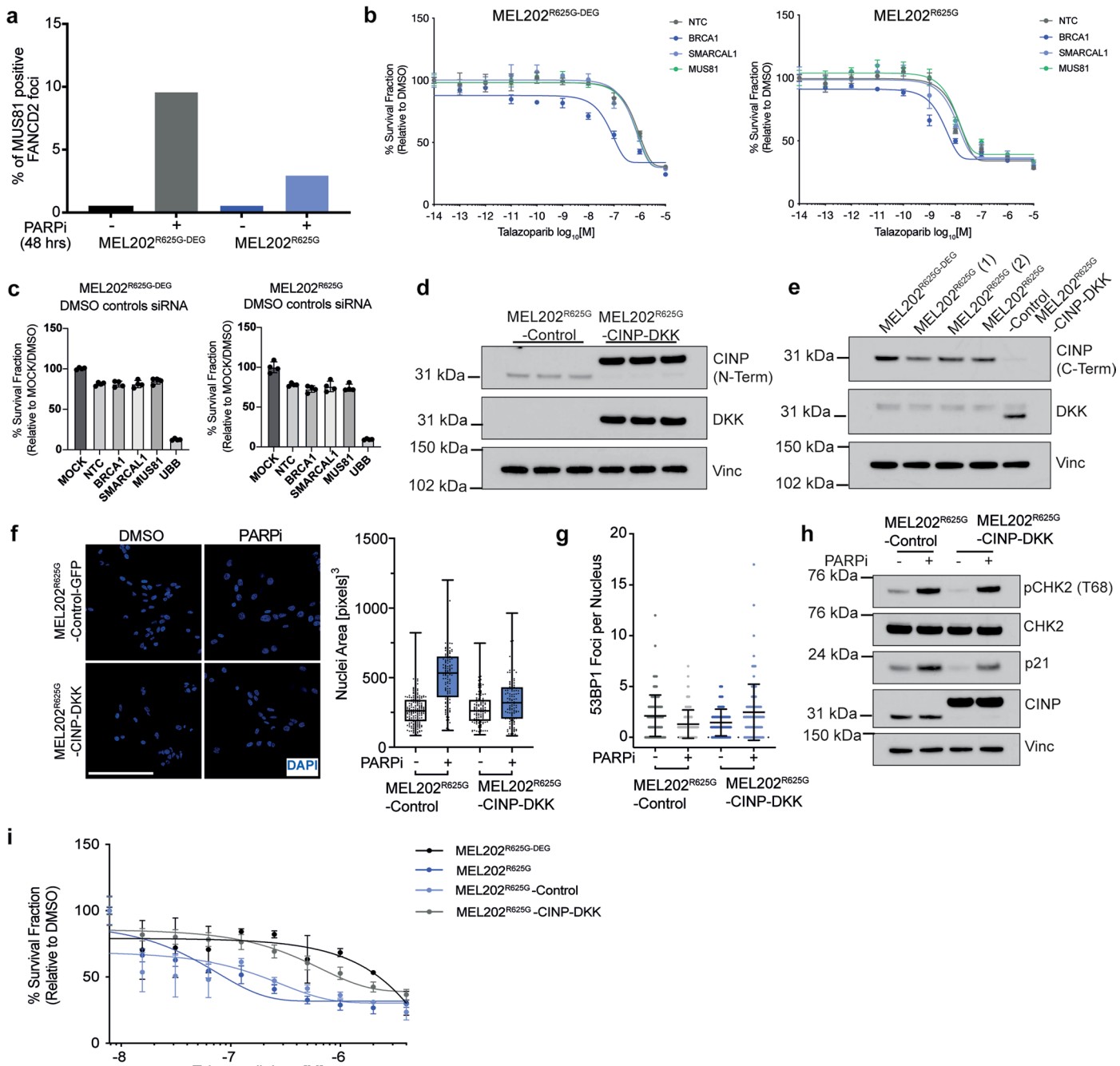

**Extended Data Fig. 8 | *SF3B1* mutant cells fail to resolve replication intermediates under PARPi exposure. a**, Bar plot showing percentage of MUS81 positive FANCD2 foci in MEL202 isogenic cells after 48 hours of 50 nM talazoparib exposure (*n* = 3 independent biological replicates). **b**, Dose–response of talazoparib exposure after NTC, *BRCA1*, *SMARCAL1* and *MUS81* mediated gene silencing in MEL202^R625G cells normalized to DMSO control (*n* = 1 biological replicate, error bars are ± s.d. of *n* = 4 technical replicates). **c**, Barplot showing cell survival relative to mock transfection of NTC, *BRCA1*, *SMARCAL1* and *MUS81* mediated gene silencing of MEL202^R625G-DEG and MEL202^R625G DMSO exposed cells from (b) (*n* = 1 biological replicate, error bars are ± s.d. of *n* = 4 technical replicates). **d**, Western blot showing CINP (N-terminal) and DKK tag expression in MEL202^R625G cells expressing control-GFP or CINP-DKK (*n* = 1 biological replicate). **e**, Western blot showing CINP (C-terminal) and DKK tag expression in MEL202 isogenic cells, and MEL202^R625G cells expressing control-GFP or

CINP-DKK. **f**, Representative immunofluorescence images and corresponding box and whiskers plot showing the nuclear area of MEL202^R625G cells expressing control-GFP or CINP-DKK after 48 hours of 50 nM talazoparib exposure (*n* = 1 biological replicate, error bars show minimum to maximum nuclear area of n > 125 individual nuclei assessed). Scale bar = 100 μm. **g**, Scatterplot showing the number of 53BP1 foci per nucleus in in MEL202^R625G cells expressing control-GFP or CINP-DKK after 3 hours of 500 nM talazoparib or DMSO exposure (*n* = 1 biological replicate, error bars show ± s.d. of n > 81 individual nuclei assessed). **h**, Western blot showing pCHK2 (T68), total CHK2, p21, and CINP (N-terminal) expression in MEL202^R625G cells expressing control-GFP or CINP-GFP, treated with 48 hours of 50 nM talazoparib (*n* = 1 biological replicate). **i**, Talazoparib 5-day dose-response curves of MEL202 isogenic cells, and MEL202^R625G cells expressing control-GFP or CINP-DKK (*n* = 1 biological replicate). Data are presented as mean values +/- s.d. of *n* = 4 technical replicates.

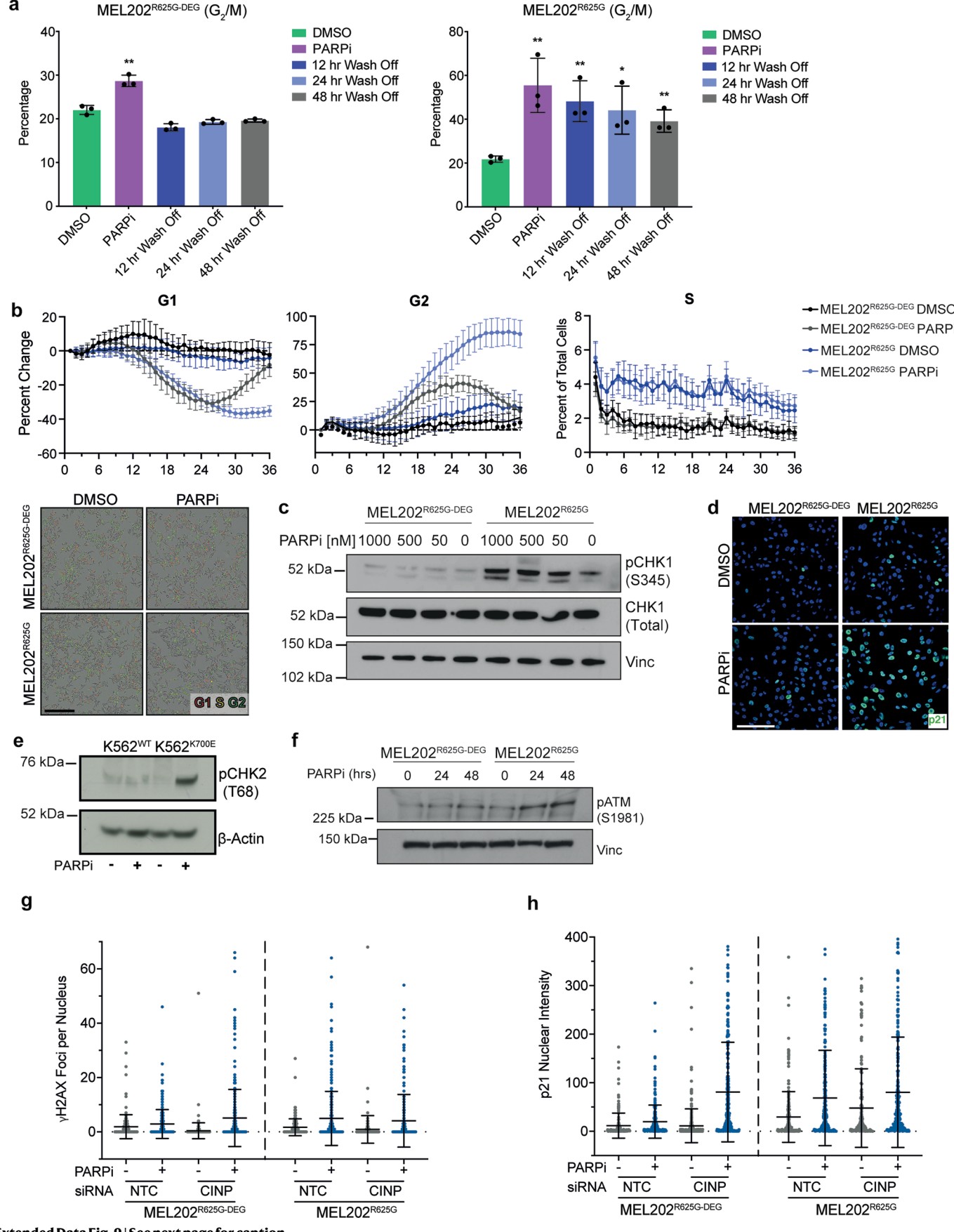

**Extended Data Fig. 9 | See next page for caption.**

**Extended Data Fig. 9 | The induction of the G$_2$/M checkpoint in *SF3B1* mutant cells. a**, Column bar graphs showing the increase in percentage of MEL202$^{R625G\text{-}DEG}$ and MEL202$^{R625G}$ isogenic cells in G$_2$/M phase after 48 hours of 50 nM talazoparib exposure, and 12, 24, and 48 hours after talazoparib removal. Data are mean of $n$ = 3 biological replicates, error bars show ± s.e.m. (unpaired two-tailed t-test (MEL202$^{R625G\text{-}DEG}$ **$P$ = 0.0022 and MEL202$^{R625G}$ **$P$ = 0.0094, **$P$ = 0.0084, *$P$ = 0.0249, **$P$ = 0.0049)). **b**, Time-course assessment of the proportion of MEL202$^{R625G\text{-}DEG}$ and MEL202$^{R625G}$ cells in each of G1 (red) and G2 (green) phase of the cell cycle over 36 hours treated with DMSO or 50 nM talazoparib plotted relative to time 0. S phase is determined by spectral overlap (red and green) and is plotted as percent of total number of cells. Representative micrographs at 36 hours are shown. Data is representative of $n$ = 2 biological replicates (Scale bar = 400 μm). **c**, Western blot of pCHK1 (S345) and total CHK1 expression in

MEL202$^{R625G\text{-}DEG}$ and MEL202 isogenic cells after 48 hours of 1000 nM, 500 nM, 50 nM, or 0 nM talazoparib exposure. **d**, Representative immunofluorescence images to corresponding Fig. 5c showing the nuclear intensity of p21 in MEL202$^{R625G}$ cells, after 48 hours of 50 nM talazoparib or DMSO exposure. **e**, Western blot of pCHK2 (T68) expression in K562 isogenic cells after 48 hours of 50 nM talazoparib or DMSO exposure. Data are representative of $n$ = 2 biological replicates. **f**, Western blot of pATM (S1981) expression in MEL202 isogenic cells after 24 or 48 hours of 50 nM talazoparib or DMSO exposure. Images are representative of two biological replicates. **g-h**, Scatter plot quantification of γH2AX (g) and p21 (h) in MEL202 isogenic cells after NTC or *CINP* gene silencing after 48 hours 50 nM talazoparib exposure. Data are of $n$ = 1 biological replicate, error bars show ± s.d.

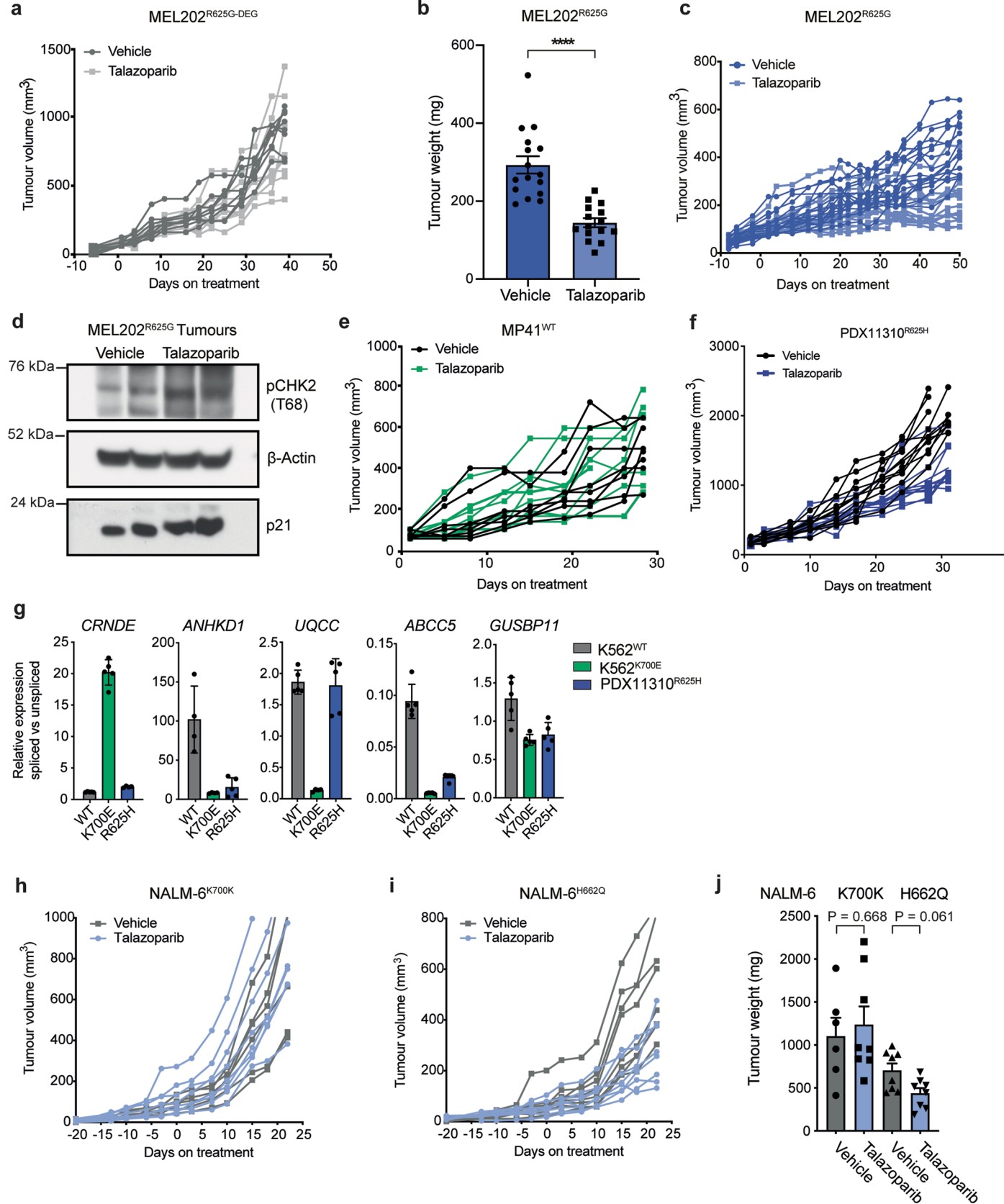

Extended Data Fig. 10 | See next page for caption.

**Extended Data Fig. 10 | PARPi suppresses *SF3B1* mutant tumor growth in vivo. a**, Chart depicting individual tumor volumes of the therapeutic response to talazoparib treatment in NSG-Nude mice bearing MEL202[R625G-DEG] xenograft tumors over time, (0.33 mg/kg). Day 0 represents the first day of treatment, (Fig. 5a). **b**, Bar plot of tumor weights from MEL202[R625G] subcutaneous tumors under treatment. At the experimental end-point, tumors were resected and weighed *ex vivo* (unpaired two-tailed t-test, ****$P < 0.0001$). **c**, Chart depicting individual tumor volumes of the therapeutic response to talazoparib treatment in NSG-Nude mice bearing *SF3B1* mutant MEL202[R625G] xenograft tumors over time, (0.33 mg/kg). Day 0 represents the first day of treatment, (Fig. 5b). **d**, Western blot of CHK2 phosphorylation at threonine 68 (pCHK2 (T68)) in two MEL202[R625G] xenograft tumors at end-point treatment with either vehicle control or talazoparib. **e**, **f**, Charts depicting individual tumor volumes of the therapeutic response to talazoparib treatment in NOD-SCID mice bearing *SF3B1*[WT] PDX MP41 (e) and SF3B1[R625H] PDX11310 patient derived xenograft (f) tumors over time, (0.33 mg/kg). Day 0 represents the first day of treatment. **g**, qRT-PCR of differentially spliced exons of indicator genes in the PDX11310 *in vivo* model. Data are mean of $n = 3$ biological replicates, error bars show ± s.e.m. **h**, **i**, Charts depicting individual tumor volumes of the therapeutic response to talazoparib treatment in CB-17 mice bearing the *SF3B1*[MUT] NALM6[H662Q] (i) and NALM6[K700K] (h) leukemia xenograft tumors over time, (0.33 mg/kg). Day 0 represents the first day of treatment. **j**, Bar plot of tumor weights from NALM-6 subcutaneous tumors under treatment, at the experimental end-point, tumors weighed ex vivo. P values shown are calculated using an unpaired two-tailed t-test.

# Reporting Summary

## Statistics

For all statistical analyses, confirm that the following items are present in the figure legend, table legend, main text, or Methods section.

| n/a | Confirmed | |
|---|---|---|
| ☐ | ☒ | The exact sample size (*n*) for each experimental group/condition, given as a discrete number and unit of measurement |
| ☐ | ☒ | A statement on whether measurements were taken from distinct samples or whether the same sample was measured repeatedly |
| ☐ | ☒ | The statistical test(s) used AND whether they are one- or two-sided *Only common tests should be described solely by name; describe more complex techniques in the Methods section.* |
| ☐ | ☒ | A description of all covariates tested |
| ☐ | ☒ | A description of any assumptions or corrections, such as tests of normality and adjustment for multiple comparisons |
| ☐ | ☒ | A full description of the statistical parameters including central tendency (e.g. means) or other basic estimates (e.g. regression coefficient) AND variation (e.g. standard deviation) or associated estimates of uncertainty (e.g. confidence intervals) |
| ☐ | ☒ | For null hypothesis testing, the test statistic (e.g. *F*, *t*, *r*) with confidence intervals, effect sizes, degrees of freedom and *P* value noted *Give P values as exact values whenever suitable.* |
| ☒ | ☐ | For Bayesian analysis, information on the choice of priors and Markov chain Monte Carlo settings |
| ☒ | ☐ | For hierarchical and complex designs, identification of the appropriate level for tests and full reporting of outcomes |
| ☐ | ☒ | Estimates of effect sizes (e.g. Cohen's *d*, Pearson's *r*), indicating how they were calculated |

*Our web collection on statistics for biologists contains articles on many of the points above.*

## Software and code

Policy information about availability of computer code

| Data collection | All histological slides were scanned at x 40 (0.25 µm/pixel) digital magnification using Hamamatsu Nanozoomer XR (Hamamatsu photonics, Hamamatsu, Japan). Digital images in .ndpi format were submitted for quantitative image analysis using HistoQuest 6.0 (Tissugnostic, Vienna, Austria) software or QPath v0.3.0. Clonogenic survival assays were quantified using MATLAB. Immunoflourescence slides were imaged on a Leica SP8 Confocal Microscope. Mitotic phase analysis of the MEL202R625G-DEG and MEL202R625G cell lines was imaged using the Zeiss Axio Observer Z1 Advanced Marianas™ Microscope attached with a CSU-W1 SoRa and quantified by eye. |
|---|---|
| Data analysis | Statistical analysis was carried out using R 3.5.0 (www.r-project.org) and GraphPad Prism 9. RNA sequencing FASTQ files were aligned to the human genome (hg38) using STAR v2.5.1b with the additional custom parameters '--twopassMode Basic --outSAMstrandField intronMotif --outSAMattributes NH HI AS nM NM XS' with transcript annotations obtained from GENCODE version 22. Differential gene expression analysis was performed using a negative binomial generalised log-linear model (glmQLFit and glmQLFTest) implemented in edgeR v3.34.0.  Normalisation factors to correct for variable sequencing depth and composition bias were calculated using the using the Trimmed Mean of M-values (TMM) method (calcNormFactors). Gene Set Enrichment Analysis was performed with FGSEA version 1.4.1 using the c2.cp.reactome gene sets obtained from the Broad Institute with the minimum pathway size set to 10. Quantification of PSI values for the alternative splicing event types (Alternative '5, Alternative '3, Exon skip, Multiple Exon Skip, Intron Retention) was performed with spladder (development version dated: 20180703) under default settings (confidence level = 3). rMATS v4.1.2 under default parameters was used as second method to identify and quantify alternative splicing events. Detection of differential alternative splicing events from both spladder  and rMATS between K562 SF3B1WT and SF3B1K700E cells was assessed by performing a differential PSI |

analysis using the limma v3.48.3. Sequence motif logos illustrating 30 bp upstream and 3bp downstream of significant alternative 3' acceptor splice sites were generated using ggseqlogo v0.1.

RNA PolII ChIP-seq data were mapped to the genome using BWA algorithm vv0.7.12 with default settings and hg38 reference genome. Only reads that passed Illumina's purity filter, aligned with no more than 2 mismatches, and mapped uniquely to the genome were used in the subsequent analysis. Peaks were called using the SICER v1.1.

Clonogenic assay NALM-6 and K562 cell lines were imaged without fixation and quantified on MATLAB vR20018b(9.5.0)

For the proteomics data, the raw files were processed with Proteome Discoverer 2.3 (Thermo Fisher) and searched using both SequestHT and Mascot (v2.3 MatrixScience) against UniProt Human Reference Proteome database (January 2018) concatenated with the cRAP contaminate sequences.

Immunofluorescence was quantified using CellProfiler (version 3.1.9). Foci were counted using the "Speckle Counting" pipeline, while phosphor-histone H3, Cajal Body, p21 and nuclear area analysis was performed using the "Cell/particle counting and scoring the percentage of stained objects" pipeline.

For manuscripts utilizing custom algorithms or software that are central to the research but not yet described in published literature, software must be made available to editors and reviewers. We strongly encourage code deposition in a community repository (e.g. GitHub). See the Nature Portfolio guidelines for submitting code & software for further information.

# Data

Policy information about availability of data

All manuscripts must include a data availability statement. This statement should provide the following information, where applicable:
- Accession codes, unique identifiers, or web links for publicly available datasets
- A description of any restrictions on data availability
- For clinical datasets or third party data, please ensure that the statement adheres to our policy

The data that support the findings of this study are available in the Supporting Information. The RNA sequencing data have been deposited in NCBI Sequence Read Archive (SRA) under accession number PRJNA849566; ChIP-seq data PRJNA968072 and the mass spectrometry proteomics data have been deposited to the ProteomeXchange Consortium via the PRIDE partner repository with the dataset identifier PXD019046.

SF3B1 mutations were collated from cBioPortal https://www.cbioportal.org/ querying MSK-IMPACT PanCancer Clinical Sequencing cohort and TCGA Pan Cancer Atlas studies. Database access 07/2020.
UniProt Human Reference Proteome database (January 2018) was used as a reference for the Mass-Spectrometry data.

# Human research participants

Policy information about studies involving human research participants and Sex and Gender in Research.

| | |
|---|---|
| Reporting on sex and gender | This is available in the original publication Pratt et al doi:10.1111/bjh.14793 |
| Population characteristics | This is available in the original publication Pratt et al doi:10.1111/bjh.14793 |
| Recruitment | This is available in the original publication Pratt et al doi:10.1111/bjh.14793 |
| Ethics oversight | This is available in the original publication Pratt et al doi:10.1111/bjh.14793 |

Note that full information on the approval of the study protocol must also be provided in the manuscript.

# Field-specific reporting

Please select the one below that is the best fit for your research. If you are not sure, read the appropriate sections before making your selection.

☒ Life sciences          ☐ Behavioural & social sciences          ☐ Ecological, evolutionary & environmental sciences

For a reference copy of the document with all sections, see nature.com/documents/nr-reporting-summary-flat.pdf

# Life sciences study design

All studies must disclose on these points even when the disclosure is negative.

| | |
|---|---|
| Sample size | Sample sizes for in vivo studies were based on the lowest number of animals required to give a probability of a type I error of 0.05 with a power of 80% assuming a 50% mean difference in drug effect between treatment and control arms in each study. This equated to 12 animals per arm of each study given a take rate of at least 70%. |

| Data exclusions | No mice were excluded |
|---|---|
| Replication | Sample sizes were sufficiently powered to enable robust reproducibility. Aside from in vivo experiments, replicate experiments were performed  in single, duplicate or triplicate independent biological replicates as  stated in the figure legends. All data was reproduced in replicate experiments. |
| Randomization | Animals were randomised when tumours reached 100mm3. All other experiments were allocated into experimental groups based on treatment (PARPi or control) and/or SF3B1 mutation status (mutant or wild-type). |
| Blinding | For all in vivo studies, the investigators were blinded to group allocations and dosing was performed by independent lab technicians. |

# Reporting for specific materials, systems and methods

We require information from authors about some types of materials, experimental systems and methods used in many studies. Here, indicate whether each material, system or method listed is relevant to your study. If you are not sure if a list item applies to your research, read the appropriate section before selecting a response.

## Materials & experimental systems

| n/a | Involved in the study |
|---|---|
| ☐ | ☒ Antibodies |
| ☐ | ☒ Eukaryotic cell lines |
| ☒ | ☐ Palaeontology and archaeology |
| ☐ | ☒ Animals and other organisms |
| ☐ | ☒ Clinical data |
| ☒ | ☐ Dual use research of concern |

## Methods

| n/a | Involved in the study |
|---|---|
| ☐ | ☒ ChIP-seq |
| ☐ | ☒ Flow cytometry |
| ☒ | ☐ MRI-based neuroimaging |

## Antibodies

| Antibodies used | Table S9. Antibodies and dilutions<br>Catalogue No. Supplier Antibody Dilution Application in this study Lot Number<br>MAB3802 Millipore Anti-53BP1 Antibody, clone BP13 1:1000 IF 3524755<br>ab180955 Abcam Anti-CINP antibody [EPR14446] ab180955 1:1000 WB, IHC GR148706-2<br>AB87913 Abcam Anti-Coilin antibody [IH10] (ab87913) 100ug 1:1000 IF GR3218582-3<br>7076S Cell Signalling Technology Anti-mouse IgG, HRP-linked Antibody #7076 1:5000 WB 36<br>05-636 Millipore Anti-phospho-Histone H2A.X (Ser139) Antibody, clone JBW301 1:1000 IF 3313712<br>7074S Cell Signalling Technology Anti-rabbit IgG, HRP-linked Antibody #7074 1:5000 WB 30<br>NA18 Millipore Anti-Replication Protein A (Ab-2) Mouse mAb (RPA34-19) 1:200 IF 3173547<br>30632S Cell Signalling Technology ATR (phospho Thr1989) antibody 1:1000 WB 1<br>2360S Cell Signalling Technology Chk1 (2G1D5) Mouse mAb #2360 1:1000 WB 3<br>2639S Cell Signalling Technology Fibrillarin (C13C3) Rabbit mAb #2639 1:1000 IF 2<br>ab133741 Abcam Lamin B1 1:1000 WB GR3244890-2<br>3873S Cell Signalling Technology Monoclonal Anti-α-Tubulin antibody produced in mouse 1:1000 WB 16<br>2947T Cell Signalling Technology p21 Waf1/Cip1 (12D1) Rabbit mAb #2947 1:1000 WB, IF 11<br>2344S Cell Signalling Technology Phospho-Chk1 (Ser317) Antibody #2344 1:1000 WB 12<br>2348S Cell Signalling Technology Phospho-Chk1 (Ser345) (133D3) Rabbit mAb #2348 1:1000 WB 18<br>2197S Cell Signalling Technology Phospho-Chk2 (Thr68) (C13C1) Rabbit mAb 1:1000 WB 12<br>53348S Cell Signalling Technology Phospho-Histone H3 (Ser10) (D7N8E) XP® Rabbit mAb #53348 1:1000 IF 1<br>ab183519 Abcam Recombinant Anti-CINP antibody [EPR14445] - N-terminal 1:1000 WB GR153682-4<br>ab133534 Abcam Recombinant Anti-Rad51 antibody [EPR4030(3)] 1:1000 IF GR219215-42<br>2808S Cell Signalling Technology Survivin (71G4B7) Rabbit mAb #2808 1:1000 WB, IF 15<br>18799S Cell Signalling Technology Vinculin (E1E9V) XP® Rabbit mAb (HRP Conjugate) 1:1000 WB 2<br>sc53382 Santa Cruz  MUS81 (MTA30 2G10/3) mouse mAb monoclonal 1/100 1:1000 IF G0721<br>NB100-182 Novus FANCD2 Rabbit polyclonal Ab 1/400 1:400 IF S-5<br>4526S Cell Signalling Technology Phospho-ATM (Ser1981) (10H11.E12) Mouse mAb 1:1000 WB 14<br>2978S Cell Signalling Technology Cyclin D1 (92G2) Rabbit mAb 1:1000 WB 13<br>2737S Cell Signalling Technology ATRIP Antbody (Rabbit) 1:1000 WB 2<br>14793S Cell Signalling Technology DYKDDDDK Tag (D6W5B) Rabbit mAb 1:1000 WB 7<br>5125S Cell Signalling Technology β-Actin (13E5) Rabbit mAb (HRP Conjugate) 1:5000 WB 6<br>ab26721 Abcam Anti-RNA polymerase II CTD repeat YSPTSPS antibody - ChIP Grade 1:1000 WB n/a<br>MABE954 Sigma-Aldrich Anti-phospho RNA Pol II (Ser5), clone 1H4B6 Antibody 1:1000 WB 3512558<br>MABE953 Sigma-Aldrich Anti-phospho RNA Pol II (Ser2), clone 3E7C7 Antibody 1:1000 WB 3692727<br>A300-996A Bethyl Laboratories SF3b155/SAP155 Polyclonal Antibody 1:1000 WB 1<br>WH0000142M1 Sigma-Aldrich Anti-PARP1 Monoclonal Antibody 1:1000 WB KC101-3G4<br>ab51052 Abcam Recombinant Anti-Hsc70 Rabbit mAb (EP1531Y) 1:1000 WB n/a<br>39097 Active Motif Anti- RNA Polymerase II (total) Mouse mAb (Clone H48) 20uL 4ug (ChIP-seq) WB 19<br>347580 BD Biosciences anti- BrdU. Mouse mAb Clone 3D4 (RUO) 1:20 IF 2077345<br>ab6326 Abcam anti- BrdU. rat mAb 1:400 IF GR3365969-8 |
|---|---|

Validation

Table S9. Antibodies and dilutions
Catalogue No. Validated in this study Validation by company
MAB3802  WB, ChIP, Flow, Flow-IC, IB, ICC/IF, IHC, IHC-Fr, IHC-P, IP, ISH, KD, KO
ab180955 KD WB, IP
AB87913  WB, ICC, IP, IHC-P, Flow, KO
7076S  WB
05-636  WB, IF, ICC, ChIP
7074S  WB
NA18  IF, IP
30632S  WB
2360S  WB, KD
2639S  WB, IF
ab133741 WB, IP, ICC/IF, IHC-P, IP, KO
3873S  WB, IHC-P, ICC/IF, Flow
2947T  WB, IP, IHC, IF, Flow, KO
2344S  WB
2348S  WB, IF, Flow
2197S  WB, IP, IHC, Flow
53348S  WB, IP, ICC/IF, ChIP, Flow
ab183519 KD WB, ICC/IF, IP
ab133534  WB, IHC-P, ICC/IF, IP, Flow
2808S  WB, IP, IHC-P, ICC/IF, Flow, KD
18799S  WB
sc53382  WB, IP
NB100-182  WB, ChIP, Flow, IB, ICC/IF, IHC, IHC-P, IP, KD, KO
4526S  WB
2978S  WB, IHC-P
2737S  WB, IF, IP
14793S  WB, IP, IHC-P, ICC/IF, Flow, ChIP
5125S  WB
ab26721  WB, IHC-P, IP, ICC/IF, ChIP
MABE954  WB, ChIP-Seq, ICC, ELISA & ChIP
MABE953  WB, ICC, ELISA, ChIP
A300-996A  WB, IP
WH0000142M1 KO WB, ELISA, IF
ab51052  IP, Flow-IC, WB, IHC-P, ICC/IF, KO
39097  WB, ChIP, ChIP-Seq
347580  Flow-IC
ab6326  ICC/IF, IHC-P, Flow-IC

Key
WB Western Blot
ChIP Chromatin Immunoprecipitation
Flow Flow cytometry
Flow-IC Flow intracellular
IB Immunobloting
ICC/IF Immunocytochemistry/Immunonblotting
IHC Immunohistochemistry
IHC-Fr Immunohistochemistry-Frozen
IHC-P Immunohistochemistry-Paraffin
IP Immunoprecipitation
ISH In situ hybridisation
KD Knock down validated
KO  Knock out validated

# Eukaryotic cell lines

Policy information about cell lines and Sex and Gender in Research

Cell line source(s)

All cell lines used in the study are derived from human.
RRID:CVCL_0004. K562 - Female parental SF3B1WT, control edited synonymous mutated SF3B1K700K and mutant
SF3B1K700E and SF3B1K666N; and RRID:CVCL_0092. NALM-6 - Male-parental SF3B1WT, control edited synonymous mutated
SF3B1K700K and mutated SF3B1H662Q, SF3B1K700E, SF3B1K666N engineered isogenic cell lines were obtained from
Horizon Discovery.
RRID:CVCL_C301. MEL202  Female parental cell line was provided by the originator Bruce Kasander Schepens Eye Research
Institute; Boston; USA
RRID:CVCL_4D13. MP46 Female patient derived xenograft  cell line was provided by the originators Fariba Nemati and Marc
Henri-Stern (Institute Curie, France)
RRID:CVCL_C302. MEL270 Male cell line was provided by the originator Bruce Kasander Schepens Eye Research Institute;
Boston; USA

| | RRID:CVCL_4D12. MP41 Female patient derived xenograft cell line was provided by the originators Fariba Nemati and Marc Henri-Stern (Institute Curie, France) |
|---|---|
| Authentication | All cell lines were authenticated using STR profiling with the Geneprint10 Kit (Promega) and were sequenced to check the retention of engineered alterations during culture |
| Mycoplasma contamination | All cell lines were tested monthly to confirm no mycoplasma infection using the MycoalertTM ®Mycoplasma Detection Kit as per manufacturer's instructions. All cell lines used in the study tested negative for mycoplasma infection. |
| Commonly misidentified lines (See ICLAC register) | No mis-identified lines were used in this study |

## Animals and other research organisms

Policy information about studies involving animals; ARRIVE guidelines recommended for reporting animal research, and Sex and Gender in Research

| Laboratory animals | 7-8 week old female CB-17 (NOD.CB17-Prkdcscid/J) , NSG-Nude (NOD.Cg-Foxn1em1Dvs Prkdcscid Il2rgtm1Wjl/J) and NOD-SCID (NOD.Cg-Prkdcscid/J) immuno-compromised mice were purchased from the Jackson Laboratory. All animals were maintained at 24-26°C ambient temperature with 55% humidity. Mice were subject to 12 hour dark-light cycles. |
|---|---|
| Wild animals | This study did not involve wild animals. |
| Reporting on sex | Female mice were used in this study |
| Field-collected samples | This study did not involve samples collected form the field. |
| Ethics oversight | The in vivo studies carried out at The Institute of Cancer Research were performed to ARRIVE guidelines and regulations as described in the UK Animals Scientific Procedures Act 1986 and according to the UK Home Office projected licences held by CJL and approved by the ethics board at The Institute of Cancer Research (maximum tumour size 15mm diameter). Additional in vivo studies were performed to local regulatory guidelines at Institut Curie (MP41 and MEL202R625G-DEG) (CEEA-IC #118, Authorization APAFiS #25870-2020060410487032-v1 given by National Authority, maximal tumour volume 2500mm3) and Crown Biosciences USA (PDX11310) (maximum tumour size 2000mm3). The maximal tumour size was not exceeded. Patients that provided samples from which PDX were generated were appropriately and fully consented. |

Note that full information on the approval of the study protocol must also be provided in the manuscript.

## Clinical data

Policy information about clinical studies
All manuscripts should comply with the ICMJE guidelines for publication of clinical research and a completed CONSORT checklist must be included with all submissions.

| Clinical trial registration | The trial is registered with ISRCTN registry (ISRCTN34386131) |
|---|---|
| Study protocol | This is available in the original publication Pratt et al doi:10.1111/bjh.14793 |
| Data collection | This is available in the original publication Pratt et al doi:10.1111/bjh.14793 |
| Outcomes | This is available in the original publication Pratt et al doi:10.1111/bjh.14793 |

## ChIP-seq

### Data deposition

☒ Confirm that both raw and final processed data have been deposited in a public database such as GEO.

☒ Confirm that you have deposited or provided access to graph files (e.g. BED files) for the called peaks.

| Data access links May remain private before publication. | For "Initial submission" or "Revised version" documents, provide reviewer access links.  For your "Final submission" document, provide a link to the deposited data. |
|---|---|
| Files in database submission | SRR24460485 PRJNA968071 SAMN34896536 Pooled_Input Pooled_Input
SRR24460487 PRJNA968071 SAMN34896534 K700K-WT_Pol2 K700K-WT_Pol2
SRR24460486 PRJNA968071 SAMN34896535 K700E-MUT_Pol2 K700E-MUT_Pol2
SRR24460488 PRJNA968071 SAMN34896533 Parental_Pol2 Parental_Pol2 |
| Genome browser session (e.g. UCSC) | No longer applicable |

## Methodology

| | |
|---|---|
| Replicates | Experiments were performed as a single replicate |
| Sequencing depth | Single end sequencing 75bp read length.<br>Total number of reads: K562WT- 35,559,728; K562K700K-39,011,608; K562K700E-37,079,970<br>Total mapped(aligned reads): K562WT- 32,001,086; K562K700K-34,594,764; K562K700E-34,072,899<br>Uniquely mapped reads: K562WT- 28,995,696; K562K700K-31,119,159; K562K700E-30,724,511 |
| Antibodies | 39097 Active Motif Anti- RNA Polymerase II (total) Mouse mAb 4ug (20uL). |
| Peak calling parameters | RNA Pol2-enriched regions were identified using the SICER algorithm v. 1.1 at a cutoff of FDR 1E-10 and a max gap parameter of 600 bp. |
| Data quality | Peaks that were on the ENCODE blacklist of known false ChIP-Seq peaks were removed. Signal maps and peak locations were used as input data to Active Motifs proprietary analysis program, which creates Excel tables containing detailed information on sample comparison, peak metrics, peak locations and gene annotations.<br><br>Filtered peaks: K562WT- 22,534; K562K700K-20,278; K562K700E-20,262 |
| Software | BWA (v0.7.12) genome alignment<br>SICER (V1.1) (peak calling)<br>bcl2fastq2 (v2.20) (processing of Illumina base-call data and demultiplexing)<br>Samtools (v0.1.19) (processing of BAM files)<br>BEDtools (v2.25.0) (processing of BED files)<br>wigToBigWig (v4) (generation of bigWIG files) |

## Flow Cytometry

### Plots

Confirm that:

☒ The axis labels state the marker and fluorochrome used (e.g. CD4-FITC).

☒ The axis scales are clearly visible. Include numbers along axes only for bottom left plot of group (a 'group' is an analysis of identical markers).

☒ All plots are contour plots with outliers or pseudocolor plots.

☒ A numerical value for number of cells or percentage (with statistics) is provided.

### Methodology

| | |
|---|---|
| Sample preparation | Cell-cycle analysis was undertaken using propidium iodide (PI) (Abcam, ab14083) and analysed on BD LSRII cell analyser. Trypsinised cells were washed twice in PBS before fixation through the dropwise addition of 70% ethanol and allowed to fix for 30 min at 4°C. Cell pellets were washed twice with PBS at 850 g and then treated with 50 ul of 100 ug/ml RNase. Finally, 200 ul of 50 ug/ml PI was used to resuspend the cell pellet ready for analysis. |
| Instrument | All samples were processed on the BD LSRII cell analyser |
| Software | FlowJO (BD biosciences) analysis software. |
| Cell population abundance | No sorting was performed. |
| Gating strategy | Forward and side scatters were set to identify single cells and doublets were excluded. Gates were then automatically set and percentages derived by use of FlowJO (BD biosciences) analysis software. |

☒ Tick this box to confirm that a figure exemplifying the gating strategy is provided in the Supplementary Information.

